# Contrasting Properties of Polymeric Nanocarriers for MRI-Guided Drug Delivery

**DOI:** 10.3390/nano13152163

**Published:** 2023-07-25

**Authors:** Natalia Łopuszyńska, Władysław P. Węglarz

**Affiliations:** Department of Magnetic Resonance Imaging, Institute of Nuclear Physics Polish Academy of Sciences, 31-342 Cracow, Poland

**Keywords:** MRI, theranostics, drug delivery, polymeric nanoparticles

## Abstract

Poor pharmacokinetics and low aqueous solubility combined with rapid clearance from the circulation of drugs result in their limited effectiveness and generally high therapeutic doses. The use of nanocarriers for drug delivery can prevent the rapid degradation of the drug, leading to its increased half-life. It can also improve the solubility and stability of drugs, advance their distribution and targeting, ensure a sustained release, and reduce drug resistance by delivering multiple therapeutic agents simultaneously. Furthermore, nanotechnology enables the combination of therapeutics with biomedical imaging agents and other treatment modalities to overcome the challenges of disease diagnosis and therapy. Such an approach is referred to as “theranostics” and aims to offer a more patient-specific approach through the observation of the distribution of contrast agents that are linked to therapeutics. The purpose of this paper is to present the recent scientific reports on polymeric nanocarriers for MRI-guided drug delivery. Polymeric nanocarriers are a very broad and versatile group of materials for drug delivery, providing high loading capacities, improved pharmacokinetics, and biocompatibility. The main focus was on the contrasting properties of proposed polymeric nanocarriers, which can be categorized into three main groups: polymeric nanocarriers (1) with relaxation-type contrast agents, (2) with chemical exchange saturation transfer (CEST) properties, and (3) with direct detection contrast agents based on fluorinated compounds. The importance of this aspect tends to be downplayed, despite its being essential for the successful design of applicable theranostic nanocarriers for image-guided drug delivery. If available, cytotoxicity and therapeutic effects were also summarized.

## 1. Introduction

To enable effective action, therapeutic agents have to be delivered to their specific destinations, which are usually the cytoplasm or nucleus of cells. However, as many drugs suffer from poor pharmacokinetics and low aqueous solubility, their effectiveness is limited, resulting in generally high therapeutic doses. Moreover, due to the small size and molecular weight of the particles, the clearance of these agents from the circulation is very rapid, leading to a short half-life that limits clinical use [1]. These issues can be overcome by providing nanosystems with high surface-to-volume ratios and distinctive physiochemical features through the integration of nanotechnology and drug design. Specifically, the use of nanocarriers for drug delivery offers the following advantages: it prevents the rapid degradation of the drug, resulting in an increased half-life of the drug in the systemic circulation; appropriate design of the nanocarrier improves the solubility and stability of drugs; it advances drug distribution and targeting of the cancer sites, ensures a sustained release of the drug, and it can reduce drug resistance by the delivery of multiple therapeutic agents [2].

Moreover, nanotechnology offers the opportunity to combine therapeutics with biomedical imaging agents and other treatment modalities to overcome the challenges of cancer diagnosis and therapy. Such an approach is referred to as “theranostics”. Initially, this term was closely related to personalized medicine, a strategy of closely linking disease diagnosis with the application of appropriate therapy. Later, “theranostic” took on a new meaning, and nowadays it is used to describe a single agent in which the diagnostic and therapeutic properties are combined. Theranostics aims to offer a more patient-specific approach through the individual adjustment of the therapeutic dose for each patient, based on the observation of the distribution of contrast agents that are linked to therapeutics. By combining both the diagnostic and therapeutic aspects, it is possible to reduce the number of agents administered to patients, their dose, and the number of invasive treatments they must undergo [3].

The most commonly used diagnostic imaging modalities are ultrasound imaging, magnetic resonance imaging (MRI), computed tomography (CT), positron emission tomography (PET), and single-photon emission computed tomography (SPECT). MRI is a technique with tomographic capabilities that does not utilize ionizing radiation to create the image. It allows for imaging at any depth within the object and has a submillimeter resolution in vivo. For theranostic purposes, MRI contrast agents are coupled with therapeutic components for the implementation of targeted therapy such as MR-guided drug delivery. There is a very wide range of contrast agents used in MRI, which are usually based on gadolinium, iron oxide, manganese oxide, or ^19^F-labelled compounds [4,5]. A more detailed description of the types and mechanisms of action of contrast agents for MRI is provided in a later section of this paper.

Anticancer treatment is an especially promising area of theranostic application. Cancer is a leading cause of death worldwide, accounting for nearly 10 million deaths in 2020 [6]. Despite rapid advancements in cancer diagnosis and treatment, there are still major obstacles to be addressed. Currently available anticancer treatment options, including radiotherapy, chemotherapy, and surgery, are invasive, painful, and sometimes imprecise, and therefore ineffective. As a result, significant acute and chronic adverse effects negatively impacting the patient’s overall outcome are observed, especially in the cases of chemotherapy and radiotherapy, which target non-specifically fast-dividing cells whether they are cancerous or not [2]. Therefore, one of the main objectives of drug discovery studies is the creation of effective therapies based on targeted platforms that can differentiate between cancer and healthy cells and deliver a therapeutic dose to cancer tissue [1].

The targeting of nanoparticles (NPs) to tumor sites can be achieved passively, actively, or through externally applied forces. Passive targeting is achieved via the enhanced permeability and retention (EPR) effect, where nanoparticle accumulation is increased due to leaky vasculature with pores of 100 nm to 2 µm in diameter and impaired lymphatic drainage in the tumor. In active targeting, a ligand linked to the theranostic nanoparticle surface interacts specifically with a receptor on the target cells. Peptides, aptamers, and antibodies are some of these ligands. Utilizing pH-sensitive, temperature-sensitive, redox-potential-sensitive, ultrasound-sensitive, and magnetic-sensitive technologies allows for targeting based on physical interactions.

The purpose of this paper is to present the recent scientific reports on polymeric nanocarriers for MRI-guided drug delivery. We selected polymeric nanocarriers as they are a very broad and versatile group of materials for drug delivery, providing high loading capacities, improved pharmacokinetics, and biocompatibility. Furthermore, these systems typically provide a significantly improved drug release profile and allow for structure modification with multiple imaging and targeting agents. Several reviews have already covered theranostic systems with MRI-trackable agents [7,8,9]; however, none of them focused on polymeric nanocarriers. While existing reviews on polymeric nanocarriers for biomedical applications [10,11,12,13,14] focus mostly on synthesis methods and the structure of synthesized constructs, none of them is dedicated to MRI-guided theranostics. A schematic illustration of the topics discussed in this review paper is presented in Figure 1. Here, we focus mostly on the contrasting properties of synthesized polymeric nanocarriers, which can be categorized into three main groups: polymeric nanocarriers (1) with relaxation-type contrast agents, (2) with chemical exchange saturation transfer (CEST) properties, and (3) with direct detection contrast agents, based on fluorinated compounds. Additionally, if available, we report their cytotoxicity and therapeutic effect.

## 2. Constituents and Morphologies of Polymeric Nanocarriers

### 2.1. The Nanocarriers Morphologies

Typically, solid colloidal particles with a diameter of less than 1000 nm are referred to as nanocarriers. However, to avoid rapid clearance after intravenous administration, prolong the circulation half-life, and at the same time increase the likelihood of crossing various biologic barriers while preventing accumulation in capillaries and healthy tissues, the most common nanoparticle size for drug delivery referred to in the literature is between 100 and 500 nm [13,15,16,17]. Depending on their internal structure, polymeric nanocarriers may be further classified as nanospheres (nanoparticles) or nanocapsules (NCs). Nanospheres are generally homogenous matrix systems in which the drug is dispersed in the material forming them. The drug can be adsorbed in their pores or less frequently at their surface, or it can be conjugated to them. Nanocapsules, on the other hand, are made up of two parts: the core and the shell. The core material can be solid, liquid, or gaseous, depending on the type of application, while the shell is formed by polymeric materials. In most cases, the drug is located in the core of the nanocapsule, while the shell protects it from the outside environment. Nevertheless, formulations with drugs incorporated in the shell or adsorbed at their surface are also possible, allowing the simultaneous delivery of one or several drugs in different nanocapsule compartments. The shell may be made permeable, semi-permeable, or impermeable, depending on the application, like the controlled release applications [13,18].

Due to their core–shell microstructure, polymeric nanocapsules have gained increased attention in recent years for use in drug delivery applications. The solid/oil core of nanocapsules can significantly improve drug loading efficiency while lowering the amount of polymeric matrix in nanoparticles when compared to polymeric nanospheres. Additionally, the polymeric shell can separate the encapsulated payload from the tissue environment, preventing the degradation or burst release caused by pH, temperature, enzymes, and other variables. Furthermore, the polymeric shell can be functionalized by smart molecules capable of interacting with specific biomolecules, allowing for targeted drug delivery [13,19,20,21].

Three morphologies based on drug incorporation mechanisms are most commonly investigated for drug delivery. First are polymeric carriers that use covalent bonds for direct drug conjugation, including linear polymers, hyperbranched polymers, and dendrimers. Dendrimers are one of the major classes of polymers. They are synthesized with a central core and monomers that branch out radially from this core in a way that resembles a tree [22,23]. The second group of nanocarriers is based on hydrophobic interactions between the cargo and nanocarriers and includes polymeric micelles from amphiphilic block copolymers [24,25]. The third group includes polymersomes, which are structures obtained by the self-assembly of amphiphilic block copolymers. As a result of their inner hydrophilic compartment, these nanostructures are more suitable for the delivery of water-soluble agents [26].

Nanoprecipitation, emulsion–diffusion, double emulsification, emulsion–coacervation, polymer-coating, and layer-by-layer (LbL) are the six classical procedures for the preparation of nanocarriers. Nonetheless, additional approaches, such as emulsion–evaporation and polymer liposome production procedures, have been employed [14] as well. We provide only a brief overview of frequently used polymers. Since several specialized reviews have already discussed formulation methods of polymeric nanocarriers, we invite readers to go directly to them for a comprehensive description of preparation methods [12,13,14,18,19]. 

### 2.2. Polymers for Nanocarrier Preparation

Polymer characteristics have a large impact on the stability, encapsulation efficiency, release profile, and biodistribution of the nanocapsule as a drug delivery vehicle. Biocompatible polymeric materials have been intensively investigated as potential compounds for the production of nanocapsules. In most cases, these polymers should be biodegradable in order to achieve the goals of payload release and nanoparticle elimination. Non-biodegradable yet biocompatible polymers such as polyethylene glycol (PEG) and polyvinyl alcohol (PVA) have also been frequently employed in the fabrication of nanoparticles. Because of their hydrophilicity, they can assist in drug release via diffusion. Furthermore, while not being degraded into smaller molecules, they might eventually be eliminated from circulation via the reticuloendothelial system [27,28]. To meet the diverse application requirements, several polymers have been used in the formulation of nanocapsules. These may be classified as natural or synthetic polymers based on their origin [18].

Peptides, proteins, nucleic acids, dextran ester, and chitosan are examples of natural polymers used. Because of their interactions with drug molecules, these molecules benefit from excellent biocompatibility but have short half-lives, non-specific distribution rates, and limited applications. Consequently, synthetic polymeric nanoparticles were proposed. The most important molecules exploited for such platforms are polylactic acid (PLA), poly(ε-caprolactone) (PCL), poly(lactide-co-glycolide) (PLGA), and poly(alkyl cyanoacrylate). Nevertheless, various other synthetic polymeric nanocarriers have also been reported, including polyaspartamide (PA), poly(L-aspartate), poly(D,L-lactic acid-co-glycolic acid), poly(ethylene glycol) (PEG), poly(N-vinyl pyrrolidone) (PVP), poly(N-isopropyl acrylamide) (PNIPAM), poly(hydroxypropyl methacrylamide) (PHPMA), poly(methyl methacrylate), poly-(chloromethyl-styrene) (PCMS), etc. [29]. Polymers that were the most frequently used for the synthesis of MRI-detectable drug delivery systems are briefly described below. For a more comprehensive description of the physiochemical and biological properties, as well as synthesis methods, we recommend the excellent reviews by S. Manandhar et al. [30], N. Larson et al. [31], M. M. Allyn et al. [32], and Y. K. Sung et al. [33].

#### 2.2.1. Poly(ε-caprolactone) (PCL)

PCL is a semi-crystalline polymer that is insoluble in alcohol and water and soluble in non-polar solvents like benzene, chloroform, and carbon tetrachloride. It is slightly soluble in acetone, DMF, ethyl acetate, and acetonitrile. Its insolubility in polar solvents is one of the major issues with the application of PCL and the synthesis of PCL-based nanocarriers [34,35]. 

On the other hand, this hydrophobic nature promotes efficient cellular uptake [36]. PCL has proven biodegradability, biocompatibility, and FDA approval for human use [37,38]. In early in vivo studies, it was observed that PCL undergoes an initial hydrolytic degradation process via ester cleavage until its molar mass is sufficiently low to allow further intracellular degradation [39,40]. The PCL degradation process involves its ultimate conversion to 6-hydroxycaproic acid, which is completely metabolized in the human body [41], which is essential for easy removal from the body after its application and makes PCL a perfect candidate for the design of drug delivery systems. Degradation of PCL is monitored by changes in molecular weight and can be tailored by its synthesis method from months to years by impacting factors like the degree of crystallization, molecular weight, and morphology [37,42]. Moreover, it has been found that PCL is excreted through urine and feces [41]. 

PCL is characterized by a melting temperature significantly above body temperature, ranging between 59 and 64 °C, and a glass transition temperature of −60 °C, and thus it maintains its semi-crystalline state in physiological temperature conditions [43]. 

Two main strategies are used for the synthesis of this polymer: polycondensation and ring-opening polymerization. Nevertheless, the green synthesis strategy has been gaining popularity among researchers to overcome toxicity issues and safety concerns [44].

#### 2.2.2. Polyethylene Glycol (PEG)

PEG is a polyether consisting of ethoxy units derived from the ring-opening polymerization of ethylene oxide. The traditional PEG is a linear polymer with chemically active hydroxyl groups at both ends [45]. PEG is biocompatible and is characterized by high water solubility [46]. It is readily cleared from the body, and it is widely used for drug conjugation. 

PEGylation is a term used to describe a popular strategy that involves the conjugation of PEG with a therapeutic agent [30]. PEGylation is known to enhance the aqueous solubility of hydrophobic drugs, prolong circulation time, minimize nonspecific uptake, and achieve specific tumor targetability through the enhanced permeability and retention effect [45,47,48]. PEGs form a hydrated PEG layer, which resists the adsorption of serum proteins and phagocytic uptake. This effect has been called a stealth effect. The stealth effect of PEGylation improves the blood circulation half-lives of biopharmaceuticals as well as nanoparticles [49].

Furthermore, PEG shows a high solubility in organic solvents and, therefore, end-group modifications are relatively easy. When attached to drugs or carriers, it provides drugs with greater physical and thermal stability as well as prevents or reduces aggregation of the drugs in vivo and during storage, as a result of the steric hindrance and/or masking of charges [50].

PEG has limited conjugation capacity since it has only one terminal functional group at the end of the polymer chain (two in the case of modified PEG). This limitation is proposed to be overcome by coupling amino acids like aspartic acids and bicarboxylic amino acids to the PEG [47]. Another limitation of PEG is that it is non-biodegradable, resulting in possible accumulation in the body if the size of the nanoparticles is greater than the renal threshold. From a theoretical point of view, a biodegradable polymer would be more beneficial in applications, since difficulties in achieving complete excretion would be avoided, although other issues, such as the toxicity of degradation products and the limited shelf life, would need to be considered [50].

PEG has been known as a safe, inert, and non-immunogenic synthetic polymer. However, PEG-related immunological issues have received considerable attention [51,52,53,54,55,56]. Anti-PEG antibodies have been found in patients who were treated with PEGylated nonhuman enzymes. Furthermore, circulating anti-PEG antibodies have been found in healthy subjects and are thought to be induced by PEG-containing cosmetics and foods [49].

#### 2.2.3. Poly-L-glutamic Acid (PGA)

PGA is made of naturally occurring l-glutamic acid connected through amide bonds as opposed to a nondegradable C-C backbone like other synthetic polymers that have been tested in clinical studies. Each repeating unit of l-glutamic acid contains a pendent-free γ-carboxyl group that is negatively charged at a neutral pH, making the polymer water soluble. The carboxyl groups also function as a means of attaching drugs. PGA is nontoxic and biodegradable which makes it a promising candidate for use as a carrier for the targeted administration of chemotherapy [57].

Electrostatic repulsion interactions between the negatively charged polymer and the relatively negatively charged surface of cells can limit its uptake by cells [58]. Nevertheless, the EPR effect, as well as the accumulation and retention of PGA–drug conjugates in solid tumors have been reported [57,59,60]. 

With increasing pH, PGA exhibits a conformational transition from a rod-like form in the α-helix state to a more random coil structure at the midway of pH 5.5, as revealed by a magnetic resonance study [61]. Therefore, PGA is expected to exist as a random coil at a neutral pH. The value of pH has a significant impact on the rate of PG’s enzymatic breakdown [62,63]. Moreover, it was discovered that both the composition and the sequential distribution of co-monomers in the copolymer chains influenced the rate of degradation [64]. Additionally, the overall biodegradation of the PG polymer can be impacted by the conjugation of therapeutic molecules to PG. Degradation of the polymer backbone may or may not result in the release of free drug, depending on the type of bonds utilized to attach the drug molecules to PGA [65].

The degradability of PGA and its derivatives has been examined in several investigations using isolated tissue lysosomal enzymes [57,66,67]. In comparison to poly(l-aspartic acid) and poly(d-glutamic acid), PGA was found to be more vulnerable to lysosomal breakdown, and the breakdown of PG results in the formation of monomeric l-glutamic acid [57].

#### 2.2.4. Poly(Lactic-co-glycolic Acid) (PLGA)

PLGA is a biocompatible polyester that is produced by a catalyzed ring-opening copolymerization of lactic acid (LA) and glycolic acid (GA) [68]. PLGA is a semicrystalline material with hydrophobic properties, and it degrades readily under physiological conditions. While PGA is a crystalline hydrophilic polymer with low water solubility and a fast degradation rate under physiological conditions, PLA is a stiff and hydrophobic polymer with low mechanical strength. As a copolymer of both, PLGA combines the intrinsic properties of its constitutional monomers where the LA/GA ratio strongly affects its degradation rate. For example, with an increase in the LA/GA ratio, the overall PLGA hydrophobicity increases, which leads to lower degradation and thus a slower drug release rate [69]. PLGA decomposes into non-toxic products (H_2_O and CO_2_) that are eliminated from the body [30]. In vivo, it degrades through hydrolysis of the ester bonds to its monomeric anions (LA and GA). While L-LA is converted into CO_2_, which is excreted through the lungs, and it is converted to pyruvate, that then enters the Krebs cycle, D-LA is not further metabolized before excretion. GA, on the other hand, is either directly excreted through the renal system or can be oxidized to glyoxylate, which is afterward further converted into glycine, serine, and pyruvate. The latter can again enter the Krebs cycle and is metabolized into CO_2_ and H_2_O [70,71,72]. 

#### 2.2.5. Poly(α-l-lysine) (PLL)

PLL is a water-soluble cationic biopolymer, built from monomeric unit α-l-lysine. Traditionally, three polymerization approaches are employed for PLL synthesis: solid-phase peptide synthesis (SPPS) [73,74], ring-opening polymerization (ROP) [75,76], and chemo-enzymatic synthesis [73]. 

PLLs use was proposed widely in various biomedical domains and the pharmaceutical field due to their inherent properties such as non-antigenicity, biocompatibility, and biodegradability. The building monomer, α-l-lysine, is one of the 20 naturally occurring amino acids. It is believed to be essential for eukaryotes and prokaryotes and plays critical roles in biological processes, including injury recovery and protein functions [73].

Under physiological conditions, PLL is positively charged due to the protonation of primary amino groups. PLL was developed as a functional biomedical material where the activity originates predominantly from this cationic property. Based on the electrostatic interactions between the positively charged PLL and the negatively charged components, PLLs have been investigated for application in nanocarrier synthesis, coating materials, and bacterial biofilm dispersal/membrane disruption [73].

On the other hand, hemolysis and cytotoxicity resulting from interactions between cationic PLL and the anionic membranes of red blood cells and vascular endothelial cells are the main concern in PLL biomedical applications. The cytotoxicity of PLL depends strongly on its molecular weight. PLLs with high molecular weight are more deleterious to both mitochondrial oxidative phosphorylation and glycolytic activity, leading to significant intracellular ATP depletion and initiating necrotic-type cell death [73,77].

PLLs are classified as hydrophilic biopolymers because of their good water solubility. The presence of alkyl groups in their side chains results in the amphiphilicity of PLLs, which is neglected. Notably, PLLs can fold into a variety of secondary structures such as α-helical, β-sheet, and random coil based on hydrogen bonding and electrostatic interactions among their backbones and side chains. This secondary structure is frequently influenced by environmental stimuli, including pH, temperature, solvent variations, and surfactants [78,79,80], and results in different hydrophobicity [73].

## 3. MR Contrast Agents and Their Mechanisms of Action

Currently available MR contrast agents (CAs) can be classified in different ways according to their various features, such as the presence or absence of metal atoms, magnetic properties, effects on the magnetic resonance image, chemical structure and ligands, or biodistribution and applications [81]. This section provides an overview of recent reports on polymeric nanocarriers for MRI-guided drug delivery, with classification based on contrast agent type. MRI CAs are classified into three major groups based on their biophysical mechanism of action and effect on MR images. These are relaxation CAs, where paramagnetic and superparamagnetic CAs can be distinguished; chemical exchange saturation transfer (CEST) CAs; and direct detection CAs.

### 3.1. Relaxation Contrast Agents 

The relaxation of water molecules surrounding the paramagnetic complex is induced by a fluctuating magnetic field generated by the Brownian motion of this complex and is described by the Solomon–Bloembergen–Morgan (SBM) Theory [82,83]. According to this theory, the observed relaxation rate of the solvent, $\frac{1}{T_{iObs}}$, is the sum of a diamagnetic term, $\frac{1}{T_{iS}}$, that corresponds to the relaxation rate of the solvent nuclei without the contrast agent, and a paramagnetic term, $\frac{1}{T_{iP}}$, which expresses the relaxation rate enhancement caused by the paramagnetic compound:(1)$$\frac{1}{T_{iObs}}=\frac{1}{T_{iS}}+\frac{1}{T_{iP}};\;i=\left(1,\;2\right).$$

Here, $T_{i},\;i=\left(1,\;2\right)$ indicate spin–lattice and spin–spin nuclear magnetic relaxation times, respectively. The paramagnetic contribution is proportional to the concentration C of the contrast agent and is described by the specific proton relaxivity, $r_{i}$, that directly refers to how efficiently the paramagnetic center enhances the relaxation rate of surrounding water protons, and thus to contrasting efficiency:(2)$$\frac{1}{T_{iObs}}=\frac{1}{T_{iS}}+r_{i}\;\cdot \;C;\;i=\left(1,\;2\right).$$

The relaxation of water protons originates from the dipole–dipole interactions between the proton nuclear spins and the fluctuating local magnetic field caused by paramagnetic center unpaired electron spins. Three contributions to the relaxation can be distinguished [82,84,85]: (1) inner-sphere relaxation, where a water ligand directly bound to the metal (water protons in the inner coordination sphere) is relaxed and transmits the relaxation effect through exchange with bulk water; (2) second-sphere relaxation, where hydrogen-bonded water molecules in the second coordination sphere, or exchangeable hydrogen atoms (such as O–H, N–H) undergo relaxation and exchange; and (3) outer-sphere relaxation, where water molecules diffusing near the complex compound can also undergo relaxation (Figure 2).

It is difficult to experimentally differentiate between second-sphere and outer-sphere water, and therefore, those groups are usually considered together as the outer-sphere water molecules. Additionally, in aqueous solutions, in most cases, the exchange of protons between the bulk water and the coordination sphere is so rapid that the observed proton relaxation rate is the weighted average of the relaxation rates in the inner and outer spheres. 

The inner sphere contribution to proton relaxivity arises from the chemical exchange of the coordinated water protons with the bulk. According to the Solomon–Bloembergen–Morgan (SBM) theory, the main factors influencing proton inner-sphere relaxivity are (1) the number of water molecules in the inner sphere, q; (2) the kinetics of water exchange between the inner sphere and the bulk, expressed by the exchange rate k_ex_ = 1/τ_m_, where τ_m_ is the mean residency time of the water ligand; (3) the rotational dynamics of the molecule, described by a rotational correlation time τ_R_; (4) the electron spin S of the complex; and (5) the electronic relaxation times [82,84,85]. Second-sphere relaxivity refers to complexes that have water molecules or exchangeable protons that have a residency time longer than the diffusion lifetime in the second coordination sphere. Therefore, its contribution to relaxation enhancement can be described by the dipolar term of equations for inner-sphere relaxivity [86]. Outer-sphere relaxivity is mainly described by translational diffusion due to the Brownian motion of free water molecules that interact with the electronic spins of the metal ions through dipolar intermolecular interactions. This process [87,88] depends on the distance of the closest approach of spins I and S (solvent protons and paramagnetic complex); $N_{A}$ is Avogadro’s number, $\left[M\right]$ is the molar concentration of the paramagnetic contrast agent, and D is the diffusion constant for relative diffusion [89].

**Figure 2 nanomaterials-13-02163-f002:**
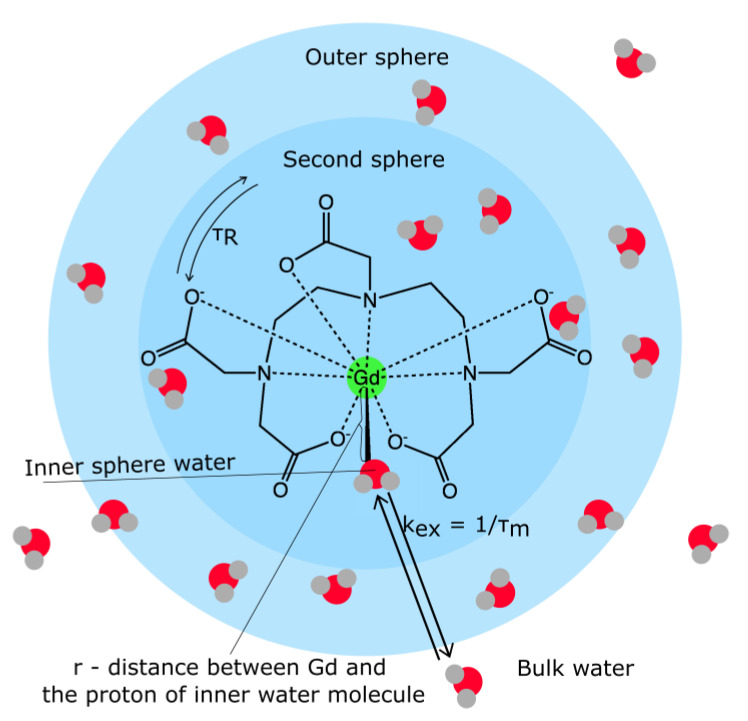
A graphical representation of the factors influencing a contrast agent’s relaxivity. (Abbreviations: see main text). Reproduced from Ref. [90] with permission from the Royal Society of Chemistry.

The T_1_ relaxation enhancement produced by the paramagnetic complex depends directly on the quantum spin number as $S\left(S+1\right)$ function and inversely on the distance between the metal ion and the proton of the water. Gadolinium (III) with S=7/2 and manganese (II) or iron (III) with S = 5/2 have been the most widely investigated as contrast agents. Due to their high toxicity and undesirable biodistribution with accumulation in the spleen, liver, and bones, Mn and Gd cannot be used in their ionic forms. For that reason, complexes with high thermodynamic and kinetic stability, such as chelates, are clinically available. Nonspecific contrast agents based on Gd(II) or Mn(II) chelates usually present similar transverse and longitudinal relaxivities; however, they are mostly used as T_1_ (positive) contrast agents [91]. The $\frac{r_{2}}{r_{1}}$ ratio for those CAs is low, below 5, which indicates that they can be successfully applied for signal amplification in T_1_-weighted images. 

The shortening of T_2_ relaxation time induced by superparamagnetic nanoparticles is determined by the translational diffusion of water molecules near the unpaired electrons. SPIOs are composed of a single magnetic domain, which is typical for nanoparticles with volumes less than the superparamagnetic limit. This occurs when it is more energetically advantageous to support the external magnetostatic energy of the single domain state rather than forming a domain structure [92]. The implications of the superparamagnetic state are as follows: in the absence of an external magnetic field, the net magnetic moment is zero; in an applied external magnetic field, superparamagnetic nanoparticles behave like paramagnets, but with much larger susceptibility and, therefore, larger saturation magnetization, which makes them efficient T_2_ CAs. For the comparison of the results obtained for proposed polymeric nanocarriers presented in the review, both the T_1_ and T_2_ relaxation agents available commercially are listed in Table 1 with their r_1_ or r_2_ molar relaxivities [92,93].

**Table 1 nanomaterials-13-02163-t001:** Examples of commercially available relaxation contrast agents.

Chemical Name	Trade Name	Mean r_1_ at 3.0 T (mM^−1^s^−1^) [94,95,96]	Mean r_2_ at 1.5 T (mM^−1^s^−1^) [97,98,99]
Gd-DTPA	Magnevist^TM^	3.3–3.7	-
Gd-DOTA	Dotarem^TM^	3.3–3.5	-
Gd-DO3A-butrol	Gadovist^TM^	4.9–5.0	-
Gd-EOB-DTPA	Primovist^TM^	5.4–6.2	-
Gd-DTPA-BMA	Omniscan^TM^	3.6–4.0	-
Gd-HP-DO3A	ProHance^TM^	3.5–3.7	-
Gd-BOPTA	MultiHence^TM^	5.1–6.3	-
Ferumoxide	Feridex^TM^	-	33–129
Ferucarbotran	Resovist^TM^	-	95–189
Ferumoxtran	Sinerem^TM^	-	65
Ferumoxytol	Faraheme^TM^	-	89

Taking into consideration the effect the contrast agent has on the MR image, relaxation CAs can be classified either as positive or negative contrast agents. Positive contrast agents shorten the T_1_, resulting in the appearance of brighter, signal-enhanced areas on T_1_-weighted images. On the other hand, negative contrast agents shorten the T_2_ relaxation time. As an effect, darker spots on T_2_-weighted images are observed [81]. Positive contrast agents include mostly the paramagnetic CAs mentioned above, as they effectively shorten T_1_. Superparamagnetic nanoparticles (mostly iron oxide NPs) are considered to be negative contrast agents. However, the r_2_/r_1_ ratio increases with the size of superparamagnetic nanoparticles, and thus ultra-small SPIONs (superparamagnetic iron oxide nanoparticles) with sizes below 10 nm can also effectively modify T_1_ relaxation and produce positive contrast with appropriate imaging sequence parameters.

### 3.2. CEST Contrast Agents

The principle of CEST imaging is very well described by its name, chemical exchange saturation transfer. Exchangeable solute protons from different molecules that resonate at a different frequency than the bulk water protons are selectively saturated using an RF pulse. This magnetic saturation is subsequently spontaneously transferred to bulk water via the chemical exchange of the excited protons with water protons. This leads to a slight attenuation of the water MR signal arising from bulk water (Figure 3). Because of the very low concentration of solute protons, which is in the range of single µM to mM, a single transfer of saturation would be insufficient to produce any noticeable change in the signal intensity of bulk water [100]. Therefore, the observable change in MR signal from water is created by the continuous transfer of excited ^1^H protons, which causes the buildup of saturation in water [101,102]. 

The CEST mechanism enhances the sensitivity of MRI, allowing the detection of low-concentration molecules indirectly, through the water signal, which makes this technique applicable to molecular and cellular imaging. The simplest classification of CEST agents includes two groups, DiaCEST and ParaCEST, and is mostly related to the chemical shift difference between the solute and bulk water. For DiaCEST, this difference usually lies in the range of 0–5 ppm [100]. DiaCEST proton exchange groups are mostly limited to amide, amine, or hydroxyl groups [103,104,105], which can be endogenous, i.e., present in the body, or exogenous, administered as a contrast agent. The biggest advantage of DiaCEST agents of the endogenous type is that CEST imaging can be performed using only modifications of the existing pulse sequences without introducing any exogenous substance that could cause adverse effects [102]. The small chemical shift difference of DiaCEST agents is, however, their biggest limitation, as it leads to partial saturation of the signal arising from a bulk water pool and also implies that very slow exchange rates are required. This can be addressed by increasing the chemical shift separation between the two exchanging pools by using exogenous ParaCEST agents. The most explored ParaCEST agents include complexes of paramagnetic lanthanide ions, such as Eu^3+^, Tm^3+^, and Yb^3+^, and paramagnetic transition metal ions: Fe, Co, and Ni. Depending on the complex, ParaCEST agents’ chemical shift values are in the range of +500 to −720 ppm with respect to the MR frequency of water [106,107]. 

In the case of ParaCEST agents, the translation to human studies may be difficult because of the technical obstacles resulting in high specific absorption rates and also due to concerns related to the toxicity of these agents. On the other hand, the application of endogenous DiaCEST [108,109] and some exogenous DiaCEST agents like nutrients, including glucose and its derivatives [110,111], to human subjects is already possible. Furthermore, some of the clinically approved CT contrast agents, including iopamidol, iopromide, iodixanol, ioversol, iobitridol, and iohexol, have been investigated as CEST agents on animal models. These agents are routinely used in CT imaging at relatively high doses, so translation to clinical CEST imaging may be possible. Specifically, this mechanism brings additional information about extracellular pH changes in the accumulation site, which is an important indicator in many diseases (for example, the Warburg effect in tumors) [112]. Another very interesting potential application of exogenous diaCEST, which is possible in human subjects, is the imaging of drug distribution. In particular, many anticancer drugs, including Gemcitabine (G), Cytarabine, Decitabine, Azacitidine, Fludarabine, Methotrexate [113], Pemetrexed [114], and Olsalazine [115], but also some anti-inflammatory (Salicylic Acid [116], Aspirin [117], and Flufenamic acid [118]), neuroprotective (Citicoline [119]), and cardiovascular drugs (Acebutolol [120]), have been investigated in the context of off-label image-guided drug delivery.

### 3.3. Direct Detection Contrast Agents

Relaxation and CEST contrast agents induce the change in local MR signal at the site of their accumulation either by shortening the relaxation rates of bulk water or by the saturation transfer of magnetization. Another group of contrast agents for MRI are CAs that can be visualized directly by the detection of a signal arising from nuclei other than hydrogen. In theory, any nuclei with a nonzero spin would produce an MR signal; however, due to factors such as the natural abundance of the isotope, relative sensitivity, and quadrupolar relaxation, only a few of them are available for magnetic resonance spectroscopy and imaging. These nuclei are ^13^C, ^23^Na, ^31^P, ^19^F, and hyperpolarized ^3^He, ^129^Xe, ^13^C, ^15^N, and ^6^Li [86]. Among the mentioned nuclei, fluorine-19 (^19^F) is perfect for in vivo imaging applications. ^19^F has an I = ½ spin, which implies that it does not undergo quadrupolar relaxation. Also, its gyromagnetic ratio γ has a value close to that of ^1^H (40.06 MHz/T for ^19^F vs. 42.58 MHz/T for ^1^H), which means that it resonates at a frequency very close to that of ^1^H ($\omega =\gamma \cdot B_{0}$) and that the standard ^1^H instruments can be used, with only minor modifications. As the MR signal is proportional to $\gamma^{3}\left(I\right)\left(I+1\right)$, and the ^19^F isotope has 100% natural abundance, its relative sensitivity is very high, equal to 0.83 (the relative sensitivity of ^1^H is 1), which is significantly higher than the relative sensitivities of other MR active nuclei, such as ^31^P (0.066), ^13^C (0.016), and ^23^Na (0.083). Last but not least, ^19^F almost does not appear physiologically in the human body. It is present at higher concentrations mostly in the bone matrix and in teeth, where it is strongly immobilized, and therefore is characterized by a very short spin–spin relaxation time and is not detectable by conventional MRI. This lack of background signal provides an excellent contrast-to-noise ratio and specificity for the exogenous ^19^F probes that can be introduced as contrast agents. However, even for ^19^F nuclei, the factor that strongly limits their application is the low signal-to-noise ratio due to the much smaller number of nuclei available for imaging than for standard ^1^H anatomical imaging, where nearly 2/3 of the nuclei present in the body contribute to the MR signal. Therefore, to increase SNR and allow detection in vivo, agents with very high ^19^F content per molecule are required to ensure sufficient concentration in the tissue [121]. Regarding the effect that direct detection contrast agents, also referred to as X-nuclei contrast agents, exert on MR images, these CAs are called “hot-spot” agents. In this case, standard ^1^H MR images as well as images of another nucleus with a different resonance frequency, for example, ^19^F, are acquired in one imaging session. Subsequently, the X-nucleus image is overlaid on the ^1^H image, where the standard MR image serves as the reference for the spatial localization of the signal arising from the X-nucleus. 

The group of chemical compounds that are most successfully used in ^19^F-MRI are perfluorocarbons (PFCs). Perfluorocarbons have a chemical structure similar to that of biologically present compounds, such as alkanes, with all hydrogen atoms substituted by fluorine [122]. The most frequently used PFCs are PFCE (perfluoro-15-crown-5-ether), which contains 20 fluorine atoms per molecule, PERFECTA (1,3bis[[1,1,1,3,3,3hexafluoro2(trifluoromethyl)propan2yl]oxy]2,2bis[[1,1,1,3,3,3hexafluoro2(trifluoromethyl)propan2yl]ox methyl]propane), with 36 chemically equivalent fluorine atoms [123], and PFOB (perfluorooctyl bromide), bearing 17 ^19^F atoms. All PFCs are hydrophobic and lipophobic; therefore, for biological applications, PFC emulsions must be stabilized with surfactants or encapsulated in polymer nanoparticles to achieve the required biocompatibility and stability. In general, PFC probes are proven to be non-toxic and biologically and chemically inert. Furthermore, they can be internalized into different cells such as macrophages, stem cells, and immune cells [124,125]. Preclinical applications of ^19^F contrast agents include mostly cell tracking [126,127,128,129], inflammation imaging in cardiac [130,131,132] and neural [133,134,135] disease models, inflammation associated with tumors [136,137], and drug delivery observations.

## 4. Polymeric Nanocarriers with MR Contrast Agents

### 4.1. Nanocarriers with Relaxation Agents

Regarding polymeric nanocarriers, two main formulations with relaxation-type contrast agents were obtained. In the first one, a single particle of contrast agent, usually in the form of SPION, forms a solid core that is covered with a polymeric shell. In the other, many molecules of contrast agents are either distributed in the polymeric matrix of the nanoparticle or are embedded or adsorbed in the shell. 

#### 4.1.1. Positive Contrast

As it was mentioned earlier in the paper, positive contrast agents are based on gadolinium chelates. The application of polymeric nanocarriers allows one to attach or embed many Gd atoms (of Gd-containing particles) in one construct, which can provide much better contrasting properties than clinically available CAs. He et al. [138] reported an intelligent biodegradable drug delivery system, Gd-DOX-HA NCPs (gadolinium-doxorubicin-loaded nanoscale coordination polymers) for targeted drug delivery and programmed drug release. The gadolinium atoms in the core of the gadolinium-based nanoscale coordination polymers (Gd NCPs) were well shielded from water, and only acidic conditions caused the decomposition of the Gd NCPs into small fragments, which facilitate the accessibility of water molecules to the gadolinium atoms, manifesting as a “turned on” T_1_ signal. More importantly, the degradation of the nanocarriers improved their biocompatibility and facilitated their clearance, presenting great potential in cancer diagnosis and treatment. The longitudinal relaxivity (r_1_) of Gd NCPs in acidic conditions was determined to be 6.58 mM^−1^s^−1^, which was 39-fold higher than that in neutral conditions. Such pH-responsive signal enhancement indicates the potential of NCPs as contrast agents for tumor positioning and the visualization of drug release. Doxorubicin delivery was also accomplished by Shalviri et al. [139] with a terpolymeric contrast agent. The polymer was synthesized by graft polymerization of methacrylic acid (MAA) and polysorbate 80 (PS 80) onto starch with multiple, chemically bound diethylenetriaminepentaacetic acid (DTPA) groups for gadolinium chelating. The addition of DOX to as-synthesized PolyGd resulted in the spontaneous formation of PolyGd-DOX nanoparticles. The r_1_ of PolyGd and PolyGd-DOX were: 21.8 ± 0.2 and 19.2 ± 0.7 mM^−1^s^−1^, respectively, at 3 T. In vivo studies showed superior and prolonged contrast enhancement compared to Omniscan^®^ at one-fourth of the equivalent dose, without adverse effects. Furthermore, the PolyGd and PolyGd-DOX accumulated in the tumor and clearly painted the tumor boundaries for at least 48 h. Similarly, a protocol for the preparation of poly(gadolinium methacrylate-co-methacrylic acid) (poly(Gd(MAA)3-co-MAA)) copolymer microspheres with high MRI contrast efficiency and controlled anti-cancer drug loading and release capability was described by Dong et al. [140] (Figure 4). Based on the electrostatic interaction between the incorporated carboxylic acid functionalities and the amine groups in DOX, Poly(Gd(MAA)3-co-MAA) could be effectively loaded with the model anti-cancer drug DOX. DOX could be released in a controlled manner under the physiological conditions of tumor cells. Moreover, the experimental results demonstrated remarkable cytotoxicity to SKOV-3 cancer cells, while possessing sufficiently low cytotoxicity to normal HEK-293T cells. The potential of poly(Gd(MAA)3-co-MAA) microspheres as an MRI contrast agent was also examined. The specific longitudinal relaxivity r_1_ of the poly(Gd(MAA)3-co-MAA) microspheres was measured to be 10.639 mM^−1^s^−1^, which was more than double the value of the clinical contrast agent Magnevist. The contrast effect was further investigated in vivo in the MR images obtained from animals before and after poly(Gd(MAA)3-co-MAA) administration, proving that prepared microspheres have excellent MRI contrast efficiency. 

Lee et al. proposed [141] (Figure 5) polymer-caged nanobins (PCNs) as a nanoscale delivery platform that can be surface-modified with targeting groups. This platform was based on a liposomal template, which allowed for the encapsulation of a high dose of small-molecule drugs (like gemcitabine) by an ion-gradient-mediated (IGM) drug-loading process. The polymer shell of the PCN contained many terminal alkyne groups on the surface; hence, azide-modified Gd III complexes could be easily conjugated to drug-loaded PCNs to result in highly effective theranostic agents. The r_1_ relaxivity of the proposed nanocarrier was 15.9 mM^−1^s^−1^. Moreover, T_1_-weighted MR images of GdIII–PCN-gemcitabine in solution were acquired at 7 T, showing a significant contrast enhancement relative to the DOTA–GdIII platform. 

Liao et al. [142] designed a multifunctional nanoscale platform that was self-assembled from a hydrophobic PLGA core and a hydrophilic paramagnetic-folate-coated PEGylated lipid (PFPL) shell. The paramagnetic DTPA-Gd chelated to the shell layer exhibited significantly higher spin–lattice relaxation (r_1_ = 14.381 mM^−1^s^−1^) than the clinically used small-molecular-weight MRI contrast agent Magnevist^®^. The PLGA core served as a nanocontainer to load and release the hydrophobic drugs (DOX). In a drug-release study, it was found that the modification of the PLGA core with a polymeric liposome shell can be a useful tool for reducing the drug-release rate. The cellular uptake of folate nanocomplex was found to be higher than that of non-folate-nanocomplex due to the folate-binding effect on the cell membrane. Finally, Szczęch et al. [143] designed novel Gadolinium (Gd)-labeled drug-loaded polyelectrolyte nanocarriers for theranostic purposes. The nanocarriers were formed via the LbL technique with biodegradable polyelectrolytes: PLL (Poly-L-lysine), PLL-Gd (Gadolinium-labelled Poly-L-lysine), and PGA (Poly-L-glutamic acid). While the anticancer drug (Paclitaxel, PTX) was encapsulated in the formed nanocarriers, the MRI contrast agent PLL-Gd constituted a part of the NCs shell. The average size of the synthesized nanocarriers was around 150 nm. MTT assays revealed that the empty gadolinium-labeled nanocarriers did not show any deleterious effects on tested cells (CT26-CEA, B16F10, 4T1, and PBMC), whereas encapsulated PTX retained its cytotoxic/cytostatic activity. The potential for the detection of Gd-labelled NCs was demonstrated using T_2_ and T_1_ NMR relaxation measurements as well as imaging with a 9.4 T preclinical MRI scanner. Depending on the core type (nanoemulsion or polymeric), the molar relaxivities r_1_ were 9.90 ± 0.60 and 8.04 ± 0.18 mM^−1^s^−1^, respectively. The obtained results showed that examined nanocapsules exhibit beneficial T_1_ relaxation properties, while enhancement of T_2_ relaxivity was not observed. Table 2 summarizes polymeric nanocarriers inducing positive MRI contrast described in this section.

#### 4.1.2. Negative Contrast

While all of the clinically approved CAs are based on Gd, the application of nanocarriers labeled with SPIONs provides additional benefits. Specifically, such theranostic agents can be additionally localized in the desired site without additional components, by the use of an external magnetic field gradient. Pan et al. [144] proposed pH-sensitive poly(β-thiopropionate) nanoparticles with a superparamagnetic core and folic acid (FA) conjugation (FA–doxorubicin–iron oxide nanoparticles FA-DOX@IONPs) to deliver an antineoplastic drug, DOX, for the treatment of folate receptor (FR)-overexpressed breast cancer. In addition to an imaging function, the nanoparticles could release their payloads in response to an environment of pH 5, such as the acidic environment found in tumors. It was demonstrated in mice xenografted with MCF-7 tumor, that when nanoparticles were entrapped with FA and/or enhanced by magnetism, significant inhibition of breast tumor growth was observed, which suggested greater antitumor effectiveness of DOX-loaded nanoparticles enhanced by FA targeting and/or magnetism than that of pure DOX. Both FA-DOX@IONPs and DOX@IONPs contained iron oxide and thus could be visualized using T_2_-weighted images. When compared with the mice before nanoparticle injection, all treatments revealed varying degrees of rounded dark areas in the tumor, denoting the aggregation of magnetic IONPs at the tumor. Moreover, the FA-DOX@IONPs plus magnetic treatment group displayed the strongest darkening at the tumor site, demonstrating the synergic effect of both targeting strategies. Szczepanowicz et al. [145] developed the method of synthesis of polyelectrolyte nanocapsules containing iron oxide nanoparticles as an MRI visible drug delivery system. Those nanocapsules were prepared by the encapsulation of nanoemulsion droplets in a hybrid multilayer shell formed by the LbL deposition of biocompatible polyelectrolytes, PGA, PLL, and Fe_2_O_3_ nanoparticles. The surface of nanocapsules was tailored for biomedical applications by the adsorption of PGA-g-PEG. Regarding magnetic resonance contrasting properties, the tested nanocapsules with two layers of Fe_2_O_3_ in the shell (AOT/PLL/PGA/Fe_2_O_3_/PGA/Fe_2_O_3_/PGA/PLL/PGA-g-PEG) displayed beneficial T_2_ relaxation properties over the pure Fe_2_O_3_ suspension, at the same concentration of iron oxide. This was attributed to the combined effect of the presence of ferromagnetic Fe_2_O_3,_ embedded in the macromolecular structure, causing perturbation of the local magnetic field as well as the interaction of the water with the macromolecular structure of nanocapsules. The presented LbL method for nanocapsule preparation was further exploited by Szczęch et al. [146] for the preparation of multifunctional magnetically responsive polymeric-based nanocarriers optimized for biomedical applications. This hybrid delivery system was composed of drug-loaded polymer PCL nanoparticles coated with a multilayer shell of bio-acceptable components: PGA and SPIONs. For the formation of PCL nanocarriers with a model anticancer drug PTX, the spontaneous emulsification solvent evaporation (SESE) method was used. Subsequently, the magnetically responsive multilayer shell was formed via the LbL method. The obtained magnetically responsive polycaprolactone nanocarriers (MN-PCL NCs) had an average size of about 120 nm. The MRI contrasting properties provided by SPIONs were evaluated using the 9.4 T preclinical magnetic resonance imaging scanner. It was confirmed that the obtained MN-PCL NCs can be successfully used as an MRI-detectable drug delivery system, with an r_2_ value of 850.1 (±10.1) mM^−1^s^−1^. The magnetic hyperthermia effect of the MN-PCL NCs was demonstrated by applying a 25 mT radio frequency (f = 429 kHz) alternating magnetic field. The specific absorption rate (SAR) that could be achieved was equal to 55 W g^−1^. Mosafer et al. [147] proposed SPIONs/DOX co-loaded PLGA-based nanoparticles targeted with AS1411 aptamer (Apt) against murine C26 colon carcinoma cells. These were developed via a modified multiple emulsion solvent evaporation method for theranostic purposes. It was shown that synthesized PLGA/DOX/SPION nanoparticles demonstrate increased cytotoxicity compared to free DOX. Moreover, active targeting with Apt offers a significant boost of cytotoxicity of about 26% compared to PLGA/DOX/SPION, while the lack of cellular cytotoxicity was confirmed for the DOX-free formulation. MR contrasting efficiency was evaluated in vivo by the MRI of BALB/c mice bearing C26 colon carcinoma tumors at 1–6 h post-injection of NPs and Apt-NPs. Very high darkening in both the tumor and liver area was observed. Moreover, a significant difference in the maximum accumulation of nanoparticles at the tumor site between mice who were administered NPs and Apt-NPs was reported, confirming the strong accumulation and penetration of the proposed formulations to the tumor through both the EPR effect as well as active tumor targeting. Luque-Michel et al. [148] developed a theranostic surfactant-coated polymer, PLGA, nanoplatform encapsulating DOX and SPIONs. The production of four types of SPION–DOX theranostic hybrid polymeric NPs (HPNPs) using different types of surfactants (T80, Brij-35, Pluronic F68, and Vitamin E-TPGS) was studied considering the importance of increasing the BBB passage by inhibiting the action of the glycoprotein P. Regarding cytotoxicity, SPION–DOX HPNPs inhibited cell viability in a dose-dependent manner and the type of surfactant used to form the HPNPs did not influence the cytotoxic effect. While similar EC50s (half maximal effective concentrations) were obtained for encapsulated and non-encapsulated DOX at 72 h of treatment, both control NPs, SPION, and SPION–PLGA NPs, were found to be non-toxic, indicating that the drug alone was responsible for the observed toxicity. The MR contrasting properties of synthesized NPs were evaluated by magnetic resonance relaxivity studies. It was demonstrated that NPs would have strong negative contrasting properties, similar to commercially available contrast agents that are based on SPIONs. Specifically, the r_2_ values were 197.80, 172.09, 158.03, and 160.15 mM^−1^s^−1^ for the NPs synthesized with T80, Brij-35, Pluronic F68, or Vitamin E-TPGS, respectively. Zhang et al. [149] (Figure 6) reported PLGA NCs that incorporated several biocompatible multimodal imaging modalities: Fluram and Cyanine 7.5 (Cy7.5) as fluorescent probes in the blue and the near-infrared (NIR) wavelengths, ^89^Zr chelated with DFO as a radio imaging probe, and SPIONs as an MRI contrast agent. The cytotoxicity of synthesized NCs was evaluated by performing an MTT assay. Human brain endothelial cells (hCMEC/D3), and primary cultures of human CD34+-derived endothelial cells did not exhibit a decrease in viability, suggesting that the NCs are biocompatible for the studied range of concentrations and times. After the in vivo injection of NCs, all animals maintained their body weight within normal values for two weeks, with no significant morbidity and no potential adverse effects. At the same time, blood sample tests to monitor potential liver and pancreatic toxicity showed normal activity of these organs. Regarding MRI contrasting properties, phantom studies, as well as in vivo scanning, were conducted to confirm the MRI performance of the NCs. Very high values of transverse relaxivity (r_2_) were obtained for 6 wt% (336 mM^−1^s^−1^) as well as 3 wt% (278 mM^−1^s^−1^) loading. Compared with other clinically used SPION systems, much higher r_2_ values were obtained. This was further reflected in the representative abdominal images, where a much clearer increase in contrast of the liver was observed with a 55% drop in signal intensity in both the T_2_-weighted images and T_2_ map after NC administration.

Fluorescent iron oxide (FIO) nanoparticles and gemcitabine (G) encapsulated in PLGA nanospheres (PGFIO) conjugated with human epidermal growth factor receptor 2, (HER-2) antibody (HER-PGFIO) were prepared by Jaidev et al. [150]. HER-PGFIO showed a sustained release of gemcitabine for 11 days in PBS (pH 7.4). In vitro cytotoxicity evaluation of HER-PGFIO in 3D MIAPaCa-2 cultures showed a 50% inhibitory concentration (IC_50_) of 0.11 mg/mL. The negative contrast property of the non-targeted theranostic nanocarrier (PGFIO) was assessed by a phantom agar gel assay. The transverse relaxivity was observed to be 773 mM^−1^s^−1^. Thus, the developed theranostic nanocarrier showed higher r_2_ relaxivity than clinically approved MRI contrast agents. Moreover, subcutaneous tumor xenografts of MIAPaCa-2 in SCID mice were developed and a tumor regression study at the end of 30 days showed significant tumor regression (86 ± 3%) in the HER-PGFIO with magnetic hyperthermia (MHT) treatment group compared to the control group. In vivo MRI imaging showed enhanced contrast in the HER-PGFIO + MHT treatment group compared to the control. Therefore, the proposed HER-PGFIO could be considered an effective nanocarrier system for the treatment of pancreatic cancer. Situ et al. proposed [151] A54 peptide-functionalized PLGA-grafted dextran (A54-Dex-PLGA). AGKGTPSLETTP peptide (A54), was used as a homing peptide to synthesize A54 peptide functional Dex-PLGA (A54-Dex-PLGA) for specific targeting of the human hepatoma cell line BEL-7402. Using DOX as a model drug, SPIO as an MRI contrast agent, and a multifunctional graft micelle delivery system, A54-Dex-PLGA/DOX/SPIO was constructed for tumor diagnosis, detection, and therapy. The DOX/SPIO-loaded micelles could specifically target the BEL-7402 cell line. In vitro MRI results also proved the specific binding ability of A54-Dex-PLGA/DOX/SPIO micelles to hepatoma cell BEL-7402. Signal intensity changes with variations in Fe_3_O_4_ concentration were observed, confirming the MR contrasting properties of A54-Dex-PLGA/DOX/SPIO and Dex-PLGA/DOX/SPIO micelles. Moreover, the in vivo MRI experiments using a BEL-7402 orthotopic implantation model further validated the targeting effect of DOX/SPIO-loaded micelles. In vitro and in vivo anti-tumor activity results showed that A54-Dex-PLGA/DOX/SPIO micelles revealed better therapeutic effects compared with Dex-PLGA/DOX/SPIO micelles and reduced toxicity compared with commercial adriamycin injection. A multifunctional nanocomposite based on dextran and SPIONs was prepared by Lin et al. [152] for drug delivery and magnetic resonance imaging (MRI). Amphiphilic dextran was synthesized by grafting stearic acid (SA) chains onto the carbohydrate backbone, denoted as Dex-g-SA. Nanoscale micelles formed by the resulting amphiphilic dextran were used as carriers for the anticancer drug DOX and a cluster of Mn-SPIONs to form a Dex-g-SA/Mn-SPION/DOX nanocomposite with a diameter of about 100 nm. In vitro cytotoxicity tests showed that free anticancer drug (DOX·HCl) cytotoxicity against MCF-7 cells is higher than those of micelles and micellar drug formulations. Moreover, Mn-SPION/DOX nanocomposite-labeled cells dispersed in gelatine phantom created much darker images than the control, providing negative contrast in MRI images (at 3.0 T). The T_2_ values of the labeled cells decrease from 241.5 to 29.5 ms when the cell number increases from 5 × 10^4^ to 8 × 10^5^, and the labeled cells’ relaxation rate (1/T_2_) was significantly higher than that of the control. Wu et al. [153] proposed a drug-loaded charge-switchable nanohybrid system (HNPs–DA) triggered by the low pH value of the tumor microenvironment (pH 6.5) to enhance the uptake efficiency of NPs in cancer cells. The nanohybrid was designed to exhibit T_2_-MRI enhancement and chemotherapy ability, ascribed to the loaded magnetite nanocluster (MNC) and paclitaxel, respectively. The HNPs–DA comprised two functional components: a biocompatible amphiphilic polymer (Pluronic F127) to act as a nanovehicle for MNC and PTX after self-assembly in an aqueous solution; and a hydrophilic polymeric shell derived from stearoyl-polyethylenimine-2,3-dimethylmalefic anhydride (SC-g-PEI-DMMA). The drug release profiles showed no significant differences in the amount of drug release between pH 7.4 and pH 6.5. To examine the MRI properties of HNPs–DA, a series of T_2_-weighted images of HNPs–DA with iron concentrations ranging from 0.07 to 0.66 mM was acquired. Compared to water, a reduction in the MRI signal was observed, with an r_2_ value of 142.68 mM^−1^s^−1^. Moreover, a CCK8 assay indicated that HNPs–DA exhibited much higher cytotoxicity against HepG2 cells at pH 6.5, indicating that HNPs–DA can more effectively block cancer cell proliferation under acidic conditions via inducing cell apoptosis. Nguyen et al. [154] combined the advantages of SPIONs and polyaspartamide (PA) biopolymer in a biological construct for cancer diagnosis and therapy. In particular, as a multifunctional biopolymer, PA, was conjugated with biotin and DOX to enhance the cell-targeting and cell-impairing abilities of the bio-construct, while SPIONs were used as a contrast agent. It was demonstrated that nanocomposites with biotin mainly targeted 4T1 cells and enhanced the uptake into cancer cells. Regarding drug release, it was shown that the encapsulation of DOX in nanocomposite has a beneficial impact on its release profile. Free DOX was released much faster than DOX embedded in the particles at most pH values. The nanocomposite released DOX effectively and impaired only 4T1 cells. Moreover, it showed good uptake by tumor cells, thus significantly hindering and slowing the tumor growth rate in tumor xenograft models of 4T1 cells on BALB/c mice. Regarding MRI properties, in the 4T1 cells, the MR signal of the sample injected with the nanoparticles without biotin turned darker than that of the control, and injection with the nanocomposite containing biotin resulted in an even stronger effect, proving that the biotin group has an important role in enhancing the ability of the cancer cells to take up the nanoparticle. Shen et al. [155] proposed the encapsulation of quantum dots, superparamagnetic Fe_3_O_4_ nanocrystals, and DOX into biodegradable PLGA polymeric nanocomposites using the double emulsion solvent evaporation method, followed by coupling to the amine group of polyethyleneimine premodified with polyethylene glycol–folic acid (PEI-PEG-FA (PPF)) segments and adsorption of vascular endothelial growth factor (VEGF)-targeted small hairpin RNA (shRNA). These drug-loaded luminescent/magnetic PLGA-based hybrid nanocomposites (LDM-PLGA/PPF/VEGF shRNA) were fabricated for tumor-specific targeting, drug/gene delivery, and cancer imaging. The in vivo T_2_-weighted MRI for EMT-6 tumor-bearing mice after intratumoral injection of LDM-PLGA/PPF showed that, compared with the control group without injection, an enhanced darkening was observed in the tumor site after the injection of Fe_3_O_4_-loaded PLGA nanoparticles. Antitumor activity was tested in vivo on the subcutaneous EMT-6 tumor xenograft model. LM-PLGA/PPF/VEGF shRNA exhibited a slight/moderate tumor growth inhibition effect, suggesting that interference against the VEGF gene could suppress tumor growth in vivo. Additionally, LDM-PLGA/PPF/SC shRNA displayed considerable tumor growth inhibition due to the continuous release of DOX in tumor tissues and increased intratumoral drug accumulation. Wang et al. [156] reported a polymer-based multifunctional composite microsphere for simultaneous magnetic resonance imaging and controlled drug delivery through a facile oil-in-water emulsion solvent-evaporation method, in which PCL was used as the enveloping matrix, Fe_3_O_4_ nanoparticles as magnetic kernel, and DOX as an anti-cancer model drug. The in vitro cytotoxicity of PCL and PCL/Fe_3_O_4_-DOX microspheres was assessed using MTT assays on HeLa cells. It was shown that the PCL has a very weak cytotoxicity on HeLa cells, while the cytotoxic effect of DOX-loaded composite microspheres increased with the increases in DOX concentration and treatment time, which confirms the obvious killing effect on HeLa cells. The contrast effect of PCL/Fe_3_O_4_-DOX microspheres under serial sample concentrations was measured. The specific relaxivity (r_2_) for PCL/Fe_3_O_4_-DOX was calculated with the value of 7.3 mg/mL/s, which confirmed that the microspheres could be used as the negative contrast agent. The oleate-covered, iron oxide particles prepared either by co-precipitation or thermal decomposition methods and incorporated into PLGA nanoparticles (PLGA-Fe-NPs) to improve their biocompatibility and in vivo stability were proposed by Sheleich et al. [157] Moreover, PLGA-Fe-NPs have been loaded with paclitaxel to obtain an MFH-triggered drug release. In vitro PTX release from the nanoparticles was estimated in PBS after AMF (alternating magnetic field) exposure. The PTX released by PLGA-Fe-NPs after 30 min of AMF exposure was significantly higher than that measured on both control and on samples exposed to AMF for only 5 min. The heat produced during the MFH treatment is sufficient to destabilize the PLGA-Fe-NPs, triggering their selective drug release. The MR contrasting properties were confirmed by very high r_2_ relaxivity values of 175–300 mM^−1^s^−1^ depending on the synthesis method. Since the synthesized NCs exhibit high r_2_ values, they can be exploited as T_2_* MRI contrast agents able to report, in real-time, their in vivo distribution. Zhang et al. [158] presented PLGA NCs as a dual-modal imaging theranostic platform for magnetic targeting protein delivery; however, other types of cargo, like enzymes, microRNAs, or drugs, could be encapsulated instead. Biocompatible photoluminescent NCs were fabricated by the chemical reaction of the high quantum yield biodegradable and photoluminescent polyester (BPLP) PLGA polymer. Subsequently, the PLGA-BPLP NCs were transformed into a magnetic resonance (MR)/photoluminescence dual-modal imaging theranostic platform by SPIONs in the polymeric shell. Moreover, the NCs were functionalized with PEG, providing a hydrophilic surface that could result in an enhanced stealth effect. Viability tests did not show signs of cell toxicity at a wide range of NC concentrations up to 500 μg/mL. Only extremely high doses (1000 μg/mL) showed a significant reduction in cell viability and number. Phantom studies were conducted to confirm the MRI contrasting properties of the capsules. T_2_ maps exhibited signal decay in a concentration-dependent manner. The calculated transverse relaxivity (r_2_) values at 7 Tesla of both non-PEGylated NC1 (263 mM^−1^s^−1^) and PEGylated NC2 (237 mM^−1^s^−1^) indicated strong negative contrasting properties for MRI. 

In Ref. [159], Cui et al. combined ligand-mediated and magnetic-guided targeting and developed a dual-functional tumor-specific delivery system (Figure 7). For the ligand-mediated targeting, the human transferrin receptor-binding peptide T7 (sequence HAIYPRH) was selected. A magnetic PLGA nanoparticulate system (MNP/T7-PLGA NPs) was synthesized by a single-emulsion solvent evaporation method, by which the hydrophobic magnetic nanoparticles (MNPs) were entrapped in the PLGA NPs. This system was used for the co-delivery of paclitaxel and curcumin (CUR) via a T7-mediated, magnetic-guided dual-targeting mechanism. The water-insoluble drugs PTX and CUR were loaded into the hydrophobic cores of T7-PLGA NPs via the hydrophobic interactions. The contrasting properties of the MNP/T7-PLGA NPs were studied by using magnetic resonance imaging and relaxometry. The r_2_ relaxivity of the MNP/T7-PLGA NPs was 281.05 mM^−1^s^−1^. Moreover, the MRI signal was monitored after tail vein injection of the MNP/T7-PLGA NPs, displaying the potential of the MNP/T7-mediated, magnetic-guided dual-targeting strategy for efficient transport across the BBB and accumulation in a brain tumor. The in vivo anti-glioma efficacy was investigated using the transplanted orthotopic U87-Luc glioma model, demonstrating the great potential of the combination therapy under the dual-targeting mechanism. 

Shang et al. [160], fabricated the biostable, biocompatible, and reducible polydopamine-coated magnetic nanoparticles (SPIONs@PDA) for both MRI diagnosis and anticancer drug delivery. The shell of the magnetic nanoparticle was a crosslinked reducible polydopamine (PDA) synthesized with N, N-bis(acrylate) cystamine (BACy), dopamine methacrylamide (DMA), and PEG methyl ether methacrylate, where BACy served as a crosslinker, PEG provided colloidal stability, and DOPA moiety as an anchor to immobilize SPIONs. In vitro cell assays were performed to evaluate the cellular uptake and intracellular drug release of DOX-loaded SPIONs@PDA, and the cytotoxicity of free DOX, blank SPIONs@PDA, and DOX-loaded SPIONs@PDA. These experimental results suggested that it took some time for SPIONs@PDA to release the most loaded drugs into the cells. Moreover, DOX-loaded SPIONs@PDA showed similar cytotoxicity to free DOX, while the blank SPIONs@PDA control showed almost no cytotoxic effect on the viability of HeLa cells. The MR contrasting properties of synthesized nano constructs were evaluated utilizing MR relaxivity measurements. The DOX-loaded SPIONs@PDA showed effective T_2_ contrast under a T_2_-weighted imaging sequence with a calculated r_2_ value of 33.53 mM^−1^s^−1^. Table 3 summarizes the polymeric nanocarriers inducing negative MRI contrast described in this section.

### 4.2. Nanocarriers with Chemical Exchange Saturation Transfer CAs

While CEST agents are being extensively developed [161,162,163] and many liposomal drug carriers have been proposed [164,165,166], only a few reports on polymeric drug delivery systems using this contrast mechanism have been reported within the last few years (Table 4). One example of such polymeric contrast was proposed by Jia et al. in [167], where the authors synthesized an acylamino-containing amphiphilic block copolymer (polyethylene glycol-polyacrylamide-poly acetonitrile, PEG-b-P(AM-co-AN)) by reversible addition–fragmentation chain transfer (RAFT) polymerization. Subsequently, based on the PEG-b-P(AM-co-AN) copolymer, a new nanomedicine PEG-PAM-PAN@DOX was constructed by nanoprecipitation (Figure 8). The authors demonstrated that the cytotoxicity of PEG-PAM-PAN@DOX was lower compared to free doxorubicin. Moreover, the polymer (PEG-PAM-PAN) without the drug (DOX) exhibited excellent biocompatibility, which made it a perfect nanocarrier to load a variety of hydrophobic small-molecule drugs for tumor chemotherapy. Also, the nanomedicine more efficiently entered the cytoplasm and nucleus of cancer cells. As the PEG-b-P(AM-co-AN) has chemical exchange saturation transfer (CEST) effects, it was possible to use CEST imaging for monitoring nanocarrier accumulation. Specifically, in vivo animal experiments showed excellent sensitivity of the CEST effect for monitoring drug accumulation (at about 0.5 ppm) in tumor areas with the post-injection CEST signal significantly higher than the pre-injection one (2.17  ±  0.88% vs. 0.09  ±  0.75%, *p*  <  0.01). Most importantly, the proposed nanocarrier was effective against breast cancer. Choi et al. [168] proposed biodegradable 3D porous poly(propylene fumarate) (PPF) scaffolds loaded with DOX that can be used for sustained drug release. The surface of the PPF scaffolds was modified with three different contrast agents for MRI. First, nanoparticles of iron oxide and manganese oxide carrying anti-cancer drugs were absorbed or mixed with the scaffold. Using nanoparticles as a drug carrier allowed for more efficient loading of drug molecules onto the PPF scaffold and facilitated monitoring of the release of drug–nanoparticle complexes from the PPF scaffold surface via changes in MRI contrast. A slow (hours to days) and functional release of the drug molecules into the surrounding solution was observed. Subsequently, the release properties of proteins and polypeptides were tested by using protamine sulfate, a chemical exchange saturation transfer (CEST) MR contrast agent, attached to the scaffold. Protamine sulfate showed a steady release rate for the first 24 h. Last, in [169], Zhang et al. proposed ionizable tertiary amine-based block copolymer PEG114-b-PDPA116 (114 and 116 refer to the numbers of repeating units in the PEG and PDPA segments, respectively) that could be used as a pH-sensitive CEST agent. It was shown that the CEST signal is not detectable when the polymers form micelles near physiological pH, as the micellar form does not contain exchangeable protons. Only when the micelles dissociate in an acidic environment is the CEST signal “turned on” due to the presence of protonated unimers having exchangeable protons. The micelle transition pH was measured by dynamic light scattering and occurred at a pH of ca. 5.9. Moreover, the zeta potential increased dramatically as the pH approached ∼5.9, which was an indication of the increased protonation of the block copolymer during the transition from micelles to unimers. Despite the lack of data on the encapsulation and release of a model drug, the proposed pH-activable micelle platform might find useful applications for in vivo MRI molecular monitoring of pH-responsive drug delivery.

### 4.3. Nanocarriers with Direct Detection CAs

Due to simultaneous hydrophobic and lipophobic characteristics (given their highly fluorinated nature), most PFCs are insoluble in any media, which is a limiting factor for their application in polymeric nanocarrier synthesis. One of the methods to overcome this problem is the encapsulation of PFCs inside a biocompatible nanocapsule that enhances their biopharmaceutical capabilities. However, these formulations are accomplished by the preparation of nano- or microemulsions stabilized by surfactants, whose use may also affect cellular uptake [170]. In this sense, the use of other fluorinated compounds, like polymers, that can constitute NC shells could be more beneficial for constructs dedicated to drug delivery. Such an approach limits the interaction between therapeutic and imaging agents and increases the achievable loading of nanocapsules with a therapeutic agent. 

Despite these challenges, several approaches for the synthesis of ^19^F-MRI-detectable polymeric nanocarriers can be mentioned (Table 5). For instance, Boissenot et al. [171] optimized the encapsulation of PTX into core–shell nanocapsules made of a PLGA-PEG shell and PFOB core to serve as theranostic agents. In in vitro tests, fabricated nanocapsules induced similar mortality of CT-26 colon cancer cells as free PTX. Additionally, in vivo, ^19^F-MRI showed that the encapsulation of PTX does not limit its ability to accumulate passively in CT-26 tumors in mice by the EPR effect. Moreover, a twofold reduction in tumor growth as compared with a negative control and a free PTX group was observed. Bo et al. [172] proposed a polymer-coated liposome system. A fluorinated amphiphile-based ^19^F-MRI-traceable liposomal drug delivery system for the in vivo tracking of DOX at a therapeutic dose was synthesized. The fluorinated amphiphile was designed as a dendrimer with highly fluorinated moieties that served as hydrophobic tails and as a ^19^F-MRI detectable agent, and monodisperse PEG as hydrophilic heads that enhance solubility, biocompatibility, and stability. Subsequently, this amphiphile was used for the formulation of ^19^F-MRI-traceable liposomes with encapsulated DOX. In the in vivo ^19^F-MRI-monitored DOX delivery studies on tumor-carrying nude mice, the distribution of the synthesized liposomes in the tumor was observed. Moreover, the amounts of DOX and amphiphile in tumors and kidneys were quantitatively measured. The co-localization of DOX and amphiphile in vivo was observed, proving the efficiency of liposomes in monitoring DOX in vivo at the therapeutic dose level. Another formulation for ^19^F-traceable nanocarriers was proposed in [173] by Zhu et al. The authors modified peptidic monodisperse PEG with fluorinated L-lysine side chains and a fluorescent N-terminal for a ^19^F-MRI and fluorescence dual-imaging traceable and thermo-responsive DOX-loaded polymer-coated liposome fabrication. The efficiency of NCs was investigated in a rodent xenograft model of human liver cancer HepG2 cells. Mice injected with theranostics had higher plasma DOX concentrations than mice injected with free DOX. Additionally, the half-life times of DOX and theranostics in plasma were <5 min and around 15 min, respectively. The therapeutic efficacy in vivo test showed considerable tumor growth inhibition in the groups treated with free DOX and theranostics. Moreover, tumor sizes in the theranostics-treated group were much smaller than those of the free DOX-treated group, which showed the improved therapeutic efficacy of theranostics. Kolouchova et al. [174] described diblock polymers that could be used as advanced drug delivery systems. The hydrophilic block consisted of poly(methyl-2-oxazoline) (PMeOx) and formed a hydrophilic shell around the particles, increasing their hydrophilicity and prolonging their circulation. The hydrophobic block contained poly(2,2-difluoroethylacrylamide) (PDFEA), which provided a narrow and intensive peak in the ^19^F-NMR spectrum, making it detectable by ^19^F-MRI. Additionally, this block contained hydrophobic ferrocene moieties, which could be oxidized into hydrophilic ferrocenium in ROS-rich environments, like tumors or inflammation, triggering the disassembly of the polymer particles and release of the cargo. Furthermore, polymers were nontoxic and biocompatible. The drug loading and entrapment efficiency, as well as release profiles, were tested for the selected model drug, doxorubicin. It was shown that the proposed polymer could limit drug release into healthy tissue while enabling drug release into pathological tissues, which is desirable for therapeutic purposes. Polymer detectability, safety, pharmacokinetics, and biodistribution were also studied in vivo. The polymer depot was visible on both ^1^H and ^19^F-MR images: while ^1^H MRI showed the depot as a nonspecific hypointense area, hot spot ^19^F-MRI specifically showed the site of administration. Moreover, the polymer remained at the site of administration for approximately 36 h. Neri et al. [175], using a covalent approach, designed a novel class of fluorinated PLGA co-polymers (F-PLGA) containing an increasing number of magnetically equivalent fluorine atoms. In particular, two novel compounds, F3-PLGA and F9-PLGA, were synthesized and characterized. In terms of ^19^F-NMR signal, F9-PLGA NPs were found more effective, and therefore more suitable for MRI-guided delivery purposes. Moreover, the drug loading process did not impair the ^19^F-NMR activity of F9-PLGA NPs, while a higher ability to encapsulate hydrophobic drugs compared to unmodified PLGA NPs was observed. For biocompatibility testing, F9-PLGA NPs loaded with DEX were applied on immortalized human glomerular endothelial cells (HCiGEnC) and podocytes (HCiPodo) at different concentrations, showing no cytotoxicity. The efficacy of F9-PLGA NPs in releasing bioactive Dexamethasone on damaged podocytes was also confirmed, showing a recovered healthy morphology of these cells after 48 h incubation with the loaded NPs. Szczęch et al. [176,177] presented a novel method for nanocarrier preparation as a fluorine magnetic resonance imaging (^19^F-MRI)-detectable drug delivery system (Figure 9). The nanocarriers were formed by the deposition of polyelectrolytes on nanoemulsion droplets via the LbL technique with the saturation approach. The polyelectrolyte multilayer shell was composed of Nafion, a fluorinated ionic polymer used as an ^19^F-MRI-detectable agent, and PLL. The surface of such prepared nanocarriers is further optimized for the passive targeting of tumors by the pegylation of their surface (i.e., by the adsorption of pegylated polyanion, PGA). The ^19^F-MRI-detectable hydrophobic nanocarriers with an average size of 170 nm were obtained and characterized. Regarding ^19^F-MRI properties, using the acquisition time of less than 30 min allowed for the detection of nanocarriers with SNR = 5 for as low a fluorine atom concentration as 2.8 × 10^18^. Such an SNR was enough to reasonably visualize the distribution of fluorine in the analyzed sample, confirming the possibility of detecting Nafion-based theranostic nanocapsules using ^19^F-MRI. A different approach to theranostic polymeric nanoparticle detection was proposed by Szczęch et al. [178]. In the latter study, the drug itself, 5-fluorouracil (5-FU), was also used as the contrast moiety as it contains a fluorine atom. 5-FU polyaminoacid-based core–shell nanocarriers were formed by the encapsulation of drug-loaded nanocores with polyaminoacid multilayer shells via the LbL method with the saturation technique. For LbL encapsulation, two oppositely charged polyaminoacids, PLL and PGA, were selected. Moreover, PGA-g-PEG was used to form an external layer of whole nanocarriers. The typical, saw-like, dependence of nanocarrier zeta potential on the adsorption of consecutive polyaminoacid layers was presented, confirming successful multilayer formation. Moreover, fluorine magnetic resonance imaging confirmed the possibility of the real-time observation of developed nanocarriers and drug accumulation inside the target. A reasonable SNR of 10 was achieved for an acquisition time as short as 8 min, confirming that utilizing ^19^F-MRI for 5-FU-loaded nanocarrier detection was possible in a timeframe of 30 min, which is preferable for further in vivo testing. Recently, Alhaidari et al. [179] reported the successful synthesis of a 5-Fluorouracil polymer conjugate for the MRI monitoring of drug release. To achieve that, a hyperbranched polymer (Hyperbranched Poly(N,N-dimethylacrylamide)) was covalently conjugated to a biodegradable oligopeptide with 5-FU. The authors showed that 5-FU release induces a change in ^19^F T_2_, detectable by ^19^F-NMR, that is sufficient for the differentiation between attached and released drug states. ^19^F nuclei within the 5-FU bound to the polymer experience the slow molecular motion that leads to a very short T_2_ relaxation time. This was reflected in a significantly broadened ^19^F- NMR signal since the signal line width is inversely proportional to the T_2_ relaxation time. Incubation of the 5-FU polymer conjugate with the enzyme induced the release of 5-FU, accompanied by an increase in the T_2_ relaxation times and, hence, a sharp ^19^F-NMR signal.

## 5. Conclusions

Nanotechnology makes it possible to improve the efficacy of personalized medicine by the combination of therapeutic and imaging agents in a single nanodevice. Such an approach can help to overcome unwanted differences in biodistribution appearing when both agents are introduced to a body separately. This could minimize the number of uncomfortable procedures and allow for better individual adjustment of therapeutic doses administered to each patient. Recent developments in multifunctional theranostic nanodevices are discussed in this review, with a focus on formulations based on polymeric nanocapsules or nanospheres with MRI CAs. We discussed these in the context of MRI-guided drug delivery. 

To produce nanoparticulate theranostic formulations, complicated synthesis techniques are frequently needed, which presents problems with repeatability and the translation of research into clinical application. We believe that the application of polymeric systems based on well-described and biocompatible polymers can overcome this difficulty. Furthermore, the embedding of the drug in the polymeric matrix or its encapsulation instead of the formation of a system based on chemical bonds preserves their therapeutic effect.

In this review, different designs employed to prepare theranostic agents for MRI-guided drug delivery were presented (Table 2, Table 3, Table 4 and Table 5). Such nanocarriers contained a model drug and a component responsible for their contrasting properties for magnetic resonance imaging. The proposed formulations were categorized either as nanocapsules or nanospheres depending on their structure. Taking into account the contrasting properties, nanocarriers were divided into three main groups: nanocarriers with relaxation type contrast agents (T_1_ and T_2_), with CEST CAs, and with direct detection CAs (based on ^19^F-NMR). 

Theranostic nanocarriers with relaxation-type CAs constitute the largest group. This can be attributed to the wide availability of commercial contrast agents that can be added to nanosystems, as well as to well-established MRI methodology for such agents. Regarding the assessment of the contrasting effect of nanocarriers with relaxation-type CAs, for all systems where r_1_ or r_2_ were reported, their values were much higher than those of commercially available CAs. Moreover, through the application of SPIONs, additional properties of magnetic targeting could be introduced, improving the theranostic agent distribution. 

Only a few polymeric designs employing CEST agents for drug distribution monitoring can be found in the literature. This kind of agent introduces contrast that can be switched on only when demanded. This can result in a more accurate distribution assessment, as the NMR signal is altered only when the additional excitation pulse is applied. 

Finally, in the case of theranostic nanocarriers with fluorinated compounds, a growing interest is observed, for several reasons. First of all, this type of contrast, called “hot-spot”, is much easier to assess. As fluorine almost does not occur in the human body, or it is immobilized by very strong bonds, the application of fluorinated compounds results in a perfect contrast-to-noise ratio. More importantly, an NMR signal of a ^19^F compound is captured directly (unlike other types, that introduce alteration in the NMR signals of nearby, mostly water, protons), which allows for reliable quantification. However, perfluorocarbons that have been successfully adapted for MRI applications, such as cell labeling and inflammation monitoring, are hydrophobic, which limits their application as building blocks for polymeric theranostic compounds. In most cases, the only solution is to co-encapsulate them with the drug inside the core of the nanocapsule, which can lead to an unwanted interaction with the drug. Therefore, new polymers that are modified with fluorine atoms have been proposed. Those had preferable MRI characteristics, i.e., a single, sharp peak in the NMR spectrum and appropriate ^19^F T_1_ and T_2_ values. Moreover, such polymers were “smart”, allowing for cargo release under the influence of stimuli like temperature or pH change.

## Figures and Tables

**Figure 1 nanomaterials-13-02163-f001:**
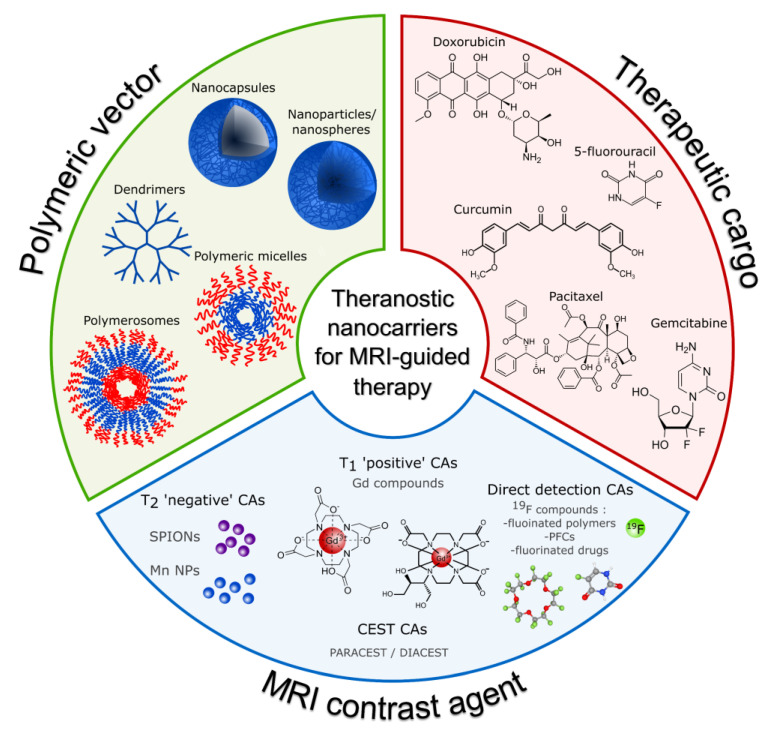
Schematic illustration of the topics discussed in this review paper, which provides an overview of recently developed tumor-targeted polymeric nanoparticles for MRI-guided drug delivery. CAs—contrast agents; SPIONs—superparamagnetic iron oxide nanoparticles; Mn NPs—manganese nanoparticles; PFCs—perfluorocarbons.

**Figure 3 nanomaterials-13-02163-f003:**
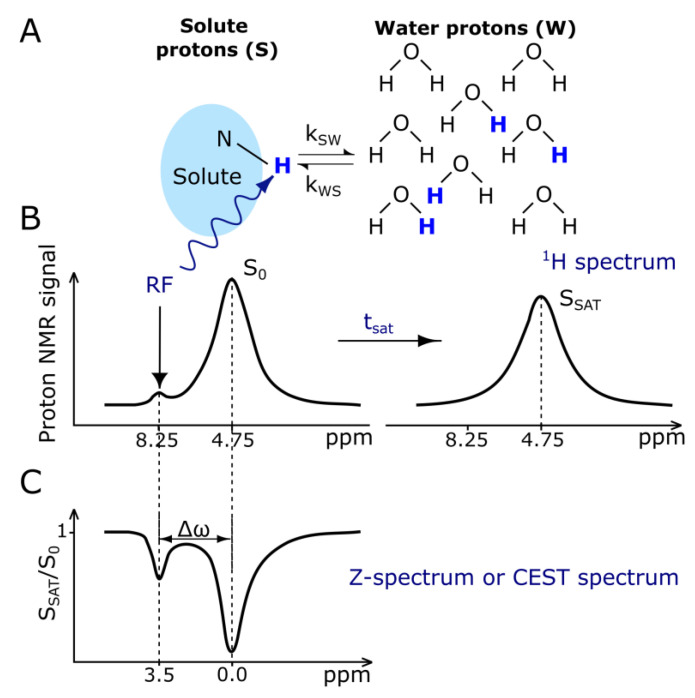
The principle of chemical exchange saturation transfer (CEST). (**A**,**B**): Solute protons (blue) are saturated at their specific resonance frequency in the proton spectrum (here 8.25 ppm for amide protons). This saturation is transferred to water (4.75 ppm) at the exchange rate ksw and nonsaturated protons (black) return. After a period (t_sat_), this effect becomes visible on the water signal (**B**, right). (**C**) Measurement of normalized water saturation (S_sat_/S_0_) as a function of irradiation frequency, generating a so-called Z-spectrum. Adapted with permission from Ref. [100], Copyright© 2011 Wiley-Liss, Inc., Hoboken, NJ, USA.

**Figure 4 nanomaterials-13-02163-f004:**
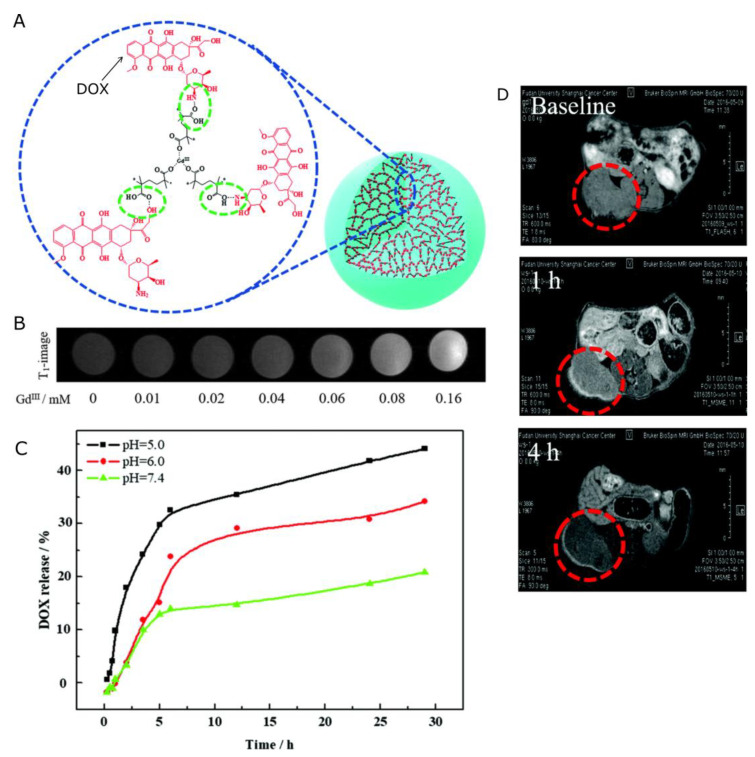
(**A**) Schematic illustration of poly(Gd(MAA)3-co-MAA) composite microspheres loaded with the anti-cancer drug DOX (red). * indicates repeated MAA monomers, in the study 1:1, 1:5, 1:10, and 1:100 ratios of Gd(MAA)_3_:MAA were investigated; (**B**) T_1_-weighted MR images for aqueous poly(Gd(MAA)3-co-MAA) microsphere dispersions, Gd(MAA)_3_:MAA = 1:10; (**C**) releasing profile of DOX from the poly(Gd(MAA)3/MAA) microspheres at pH 5.3, pH 6.0, and pH 7.4 at 37 °C (cNaCl = 0.15 M), Gd(MAA)_3_:MAA = 1:10; (**D**) in vivo MRI scans of cancer. Form the left: baseline image was acquired before the poly(Gd(MAA)3-co-MAA) injection, 1 h, and and 4 h after poly(Gd(MAA)3-co-MAA) administration, Gd(MAA)_3_:MAA = 1:10. Adapted from Ref. [140] with permission from the Centre National de la Recherche Scientifique (CNRS) and the Royal Society of Chemistry.

**Figure 5 nanomaterials-13-02163-f005:**
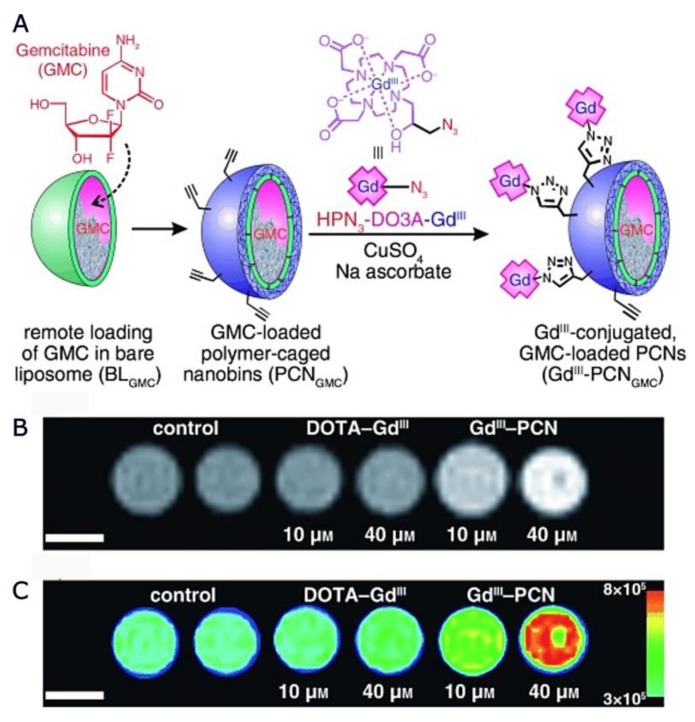
(**A**) Preparation of gemcitabine-loaded, gadolinium(III)-conjugated polymer-caged nanobins (Gd^III^–PCN_GMC_) by copper(I)-catalyzed click ligation. (**B**) T_1_-weighted MR image of NIH/3T3 mouse fibroblast cells incubated with solutions of Gd^III^–PCNs (Gd^III^ concentrations: 10 μM and 40 μM, Gd^III^/lipid = 0.39) and DOTA–Gd^III^ (Gd^III^ concentrations: 10 μM and 40 μM) for 24 h. (**C**) The corresponding image-intensity color map for the panel. The scale bars at the bottom left corners in (**B**,**C**) correspond to 1.0 mm. Adapted with permission from Ref. [141]. Copyright© 2010 WILEY-VCH Verlag GmbH & Co. KGaA, Weinheim, Germany.

**Figure 6 nanomaterials-13-02163-f006:**
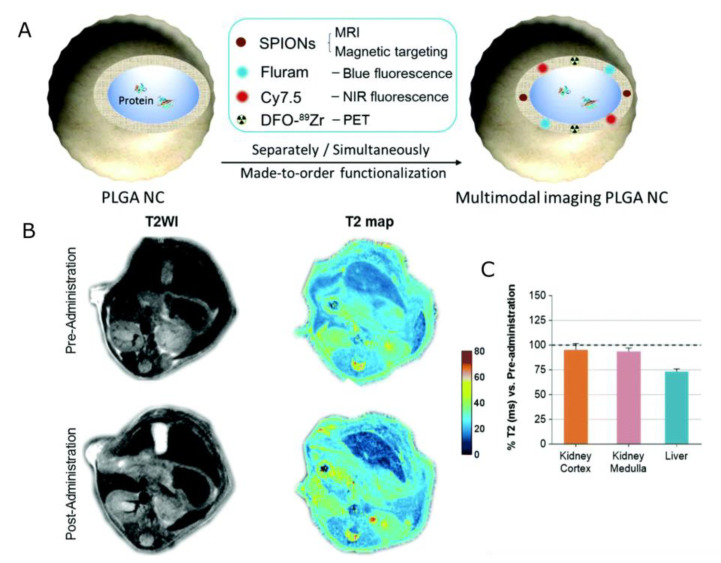
(**A**) Schematic illustration of a PLGA NC as a tailor-made multimodal theranostic platform. (**B**) In vivo mouse MRI of the PLGA–SPIONs NCs: representative T_2_-weighted images and T_2_ maps before and after NC administration. (**C**) Relaxation time (ms) calculated for each ROI corresponding to the analyzed anatomical structures. Adapted from Ref. [149] with permission from the Royal Society of Chemistry.

**Figure 7 nanomaterials-13-02163-f007:**
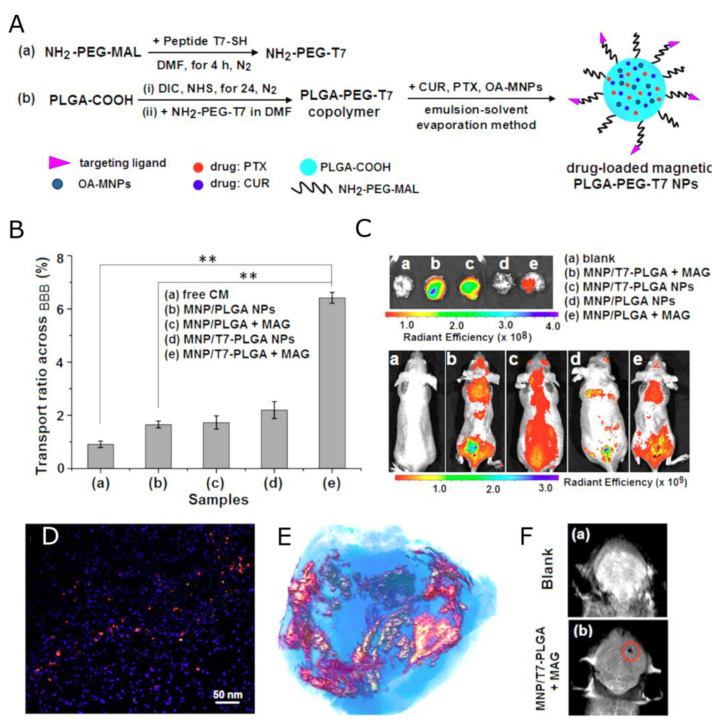
(**A**) Synthesis of PLGA-PEG-T7 polymer (a) and drug-loaded MNP/T7-PLGA NPs (b); transporting efficiency across the BBB in vitro and in vivo: (**B**) transport ratios across the in vitro BBB of different NPs at 4 h; ** *p* < 0.01; (**C**) in vivo distribution of the nanoparticle systems by IVIS animal imaging system after injection via tail vein at 4 h; (**D**) image of brain tissue section by confocal laser scanning microscope; (**E**) iron distribution in the brain examined by synchrotron radiation X-ray; (**F**) MRI of MNP/T7-PLGA NPs. The results demonstrated brain targeting mediated by dual targeting. Adapted with permission from Ref. [159]. Copyright 2016 American Chemical Society.

**Figure 8 nanomaterials-13-02163-f008:**
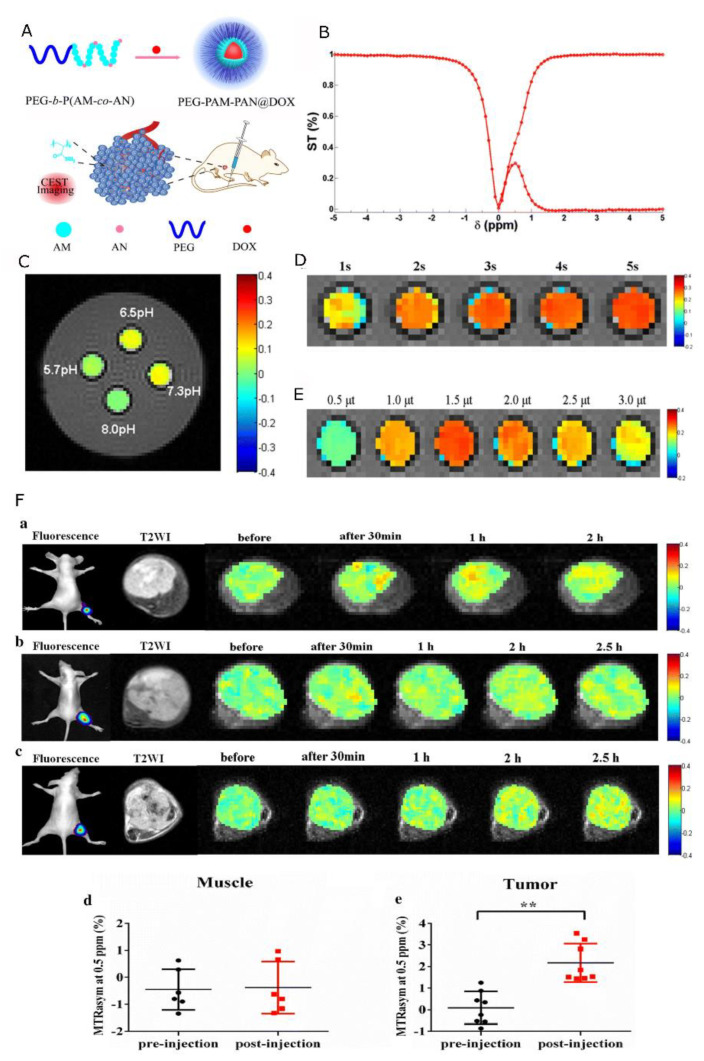
(**A**) Schematic diagram of the fabrication of PEG−PAM−PAN@DOX for chemotherapy and CEST imaging; (**B**) Z−spectra of PEG−PAM−PAN@DOX showed that the CEST effects were at approximately 0.5 ppm; (**C**) CEST imaging of PEG−PAM−PAN@DOX at different pH values; (**D**) CEST imaging of PEG−PAM−PAN@DOX at different saturation times (s) showing that the effect increase with the saturation time; (**E**) CEST imaging of PEG−PAM−PAN@DOX at different saturation power (μT) and the peak at 1.5 μT; (**F**) In vivo CEST imaging of nanomedicines intravenously injected in mice bearing MDA-MB-231 breast cancer xenografts. The imaging showed that the nanomedicine was accumulated in tumor areas and peaked at 1 h (**a**, *n* = 2), 2 h (**b**, *n* = 4), and 2.5 h (**c**, *n* = 2) after tail intravenous injection. The relative MTRasym at 0.5 ppm for the muscle and entire tumor for the two groups, respectively (**d**,**e**) (** *p* < 0.01, paired *t* test). Adapted with permission from Ref. [167]. Copyright© 2019, The Author(s).

**Figure 9 nanomaterials-13-02163-f009:**
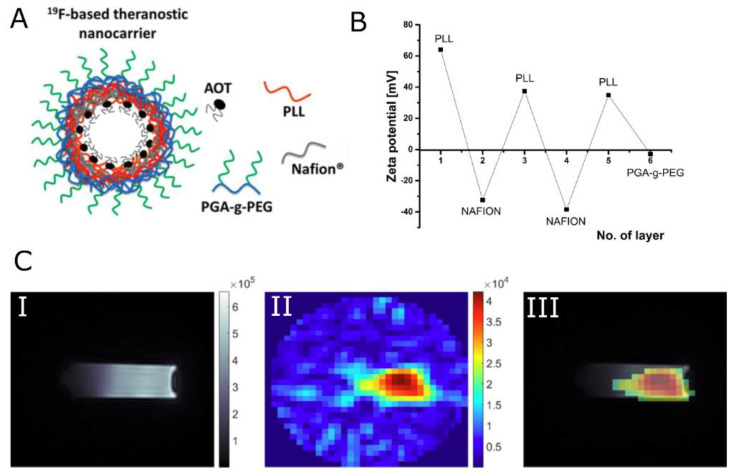
(**A**) Schematic structure of polyelectrolyte nanocapsules for ^19^F-MRI; (**B**) saw-like pattern of the dependence of nanocarrier zeta potential on the adsorption of subsequent layers; (**C**) I ^1^H MR image of phantom containing nanocarriers (UTE3D; FOV, 4.0 cm; MTX, 128; NA, 1; TA, ∼7 min), II corresponding ^19^F-MR image (UTE3D; FOV, 4.0 cm; MTX, 32; NA, 256; TA, 1 h and 48 min), and III an overlay of the ^1^H and ^19^F images. Adapted with permission from Ref. [176]. Copyright© 2020 American Chemical Society.

**Table 2 nanomaterials-13-02163-t002:** Summary of MRI-traceable polymeric nanocarriers for drug delivery applications with positive contrast agents (T_1_-reducing).

Structure	Overview	Model Drug ^1^	MRI Contrasting Properties	Ref
Nanocapsules	Acid-degradable gadolinium-doxorubicin-loaded nanoscale coordination polymer (Gd-Dox NCPs) core and hyaluronic acid shell.	DOX	r_1_ = 6.58 mM^−1^s^−1^	[138]
Nanospheres	The multifunctional terpolymeric system achieved by the polymerization of methacrylic acid and polysorbate onto starch with multiple, chemically bound DTPA groups for gadolinium chelating.	DOX	PolyGd: r_1_ = 21.8 mM^−1^s^−1^ PolyGd-DOX: r_1_ = 19.2 mM^−1^s^−1^	[139]
Nanospheres	Poly(gadolinium methacrylate-co-methacrylic acid) copolymer microspheres.	DOX	r_1_ = 10.64 mM^−1^s^−1^	[140]
Nanocapsules	Gd-loaded liposome core with a polymeric shell of PCL with azide-modified Gd III complexes conjugated to the surface.	G	r_1_ = 15.9 mM^−1^s^−1^ (at a Gd III/lipid ratio of 0.45)	[141]
Nanocapsules	Self-assembled, hydrophobic PLGA core and a hydrophilic paramagnetic-folate-coated PEGylated lipid shell with (DTPA-Gd) chelated to the shell layer.	DOX	r_1_ = 14.38 mM^−1^s^−1^	[142]
Nanocapsules	Multilayer shell of biodegradable polyelectrolytes: PLL, PLL-Gd, and PGA. Anticancer drug PTX encapsulated in the nanocarriers; the MRI contrast agent PLL-Gd constituted a part of the NCs shell.	PTX	r_1_ = 9.90 mM^−1^s^−1^ (for nanoemulsion core) r_1_ = 8.04 mM^−1^s^−1^ (for polymeric core)	[143]

^1^ DOX—doxorubicin, G—gemcitabine, PTX—Paclitaxel.

**Table 3 nanomaterials-13-02163-t003:** Summary of MRI-traceable polymeric nanocarriers for drug delivery applications with negative contrast agents (T_2_-reducing).

Structure	Overview	Model Drug ^1^	MRI Contrasting Properties	Ref
Nanocapsules	The pH-sensitive poly(β-thiopropionate) nanoparticles with a superparamagnetic core and folic acid (FA) conjugation (FA-doxorubicin-iron oxide nanoparticles (FA-DOX@IONPs)).	DOX	In vivo observation of T_2_-dependent darkening in the tumor site.	[144]
Nanocapsules	Polyelectrolyte nanocapsules with multilayer shell containing iron oxide nanoparticles as MRI visible drug delivery system.	-	Nanocapsules with two layers of Fe_2_O_3_ in the shell (AOT/PLL/PGA/Fe_2_O_3_/PGA/Fe_2_O_3_/PGA/PLL/PGA-g-PEG) displayed beneficial T_2_ − relaxation properties over the pure Fe_2_O_3_ suspension.	[145]
Nanocapsules	Drug-loaded polymer nanoparticles PCL coated with a multilayer shell of bio-acceptable components: PGA and SPIONs.	PTX	r_2_ = 850.1 mM^−1^s^−1^	[146]
Nanospheres	The SPIONs/DOX co-loaded PLGA-based nanoparticles targeted with AS1411 aptamer.	DOX	In vivo observation of T_2_-dependent darkening in tumor and liver site	[147]
Nanospheres	Surfactant-coated polymer PLGA nanoplatform co-encapsulating (DOX) and SPIONs.	DOX	r_2_ = 158.03–197.80 mM^−1^s^−1^ (depending on surfactant)	[148]
Nanocapsules	PLGA NCs with several biocompatible multimodal imaging modalities: Fluram and Cyanine 7.5 as fluorescent probes, ^89^Zr chelated with DFO as a radio imaging probe, and SPIONs as an MRI contrast agent.	proteins (BSA)	r_2_ = 336 or 278 mM^−1^s^−1^ (for higher and lower SPIONs loading, respectively)	[149]
Nanospheres	Fluorescent iron oxide nanoparticles and G-encapsulated PLGA nanospheres, conjugated with HER-PGFIO antibody.	G	r_2_ = 773 mM^−1^s^−1^	[150]
Nanocapsules	The A54 peptide-functionalized PLGA-grafted dextran (A54-Dex-PLGA) micelles with encapsulated DOX and SPIONs.	DOX	The dependences of 1/T_2_* on Fe concentration presented high slopes. Contrasting properties confirmed in vivo.	[151]
Nanocapsules	Micelles formed with Amphiphilic dextran; stearic acid (SA) chains drafted onto the carbohydrate backbone; encapsulating DOX and a cluster of Mn-SPIONs in a hydrophobic core.	DOX	T_2_ values of the labeled cells decreased from 241.5 to 29.5 ms when the cell number increases from 5 × 10^4^ to 8 × 10^5^.	[152]
Nanocapsules	Biocompatible amphiphilic polymer (Pluronic F127) self-assembled with magnetic nanocluster and PTX as the core; hydrophilic stearoyl-polyethylenimine-2,3-dimethylmalefic anhydride (SC-g-PEI-DMMA) shell.	PTX	r_2_ = 142.68 mM^−1^s^−1^	[153]
Nanospheres	Multifunctional biopolymer with PA conjugated with biotin, DOX, and SPIONs.	DOX	r_2_ not available; However, 1/T_2_ rates increased gradually vs. the SPION concentrations (in µg/mL of Fe).	[154]
Nanospheres	The encapsulation of quantum dots, SPIONS, and DOX into PLGA polymeric nanocomposites. Coupling of the amine group of polyethyleneimine premodified with PEG acid (PEI-PEG-FA (PPF)) segments and adsorption of vascular endothelial growth factor (VEGF)-targeted small hairpin RNA (shRNA).	DOX	r_2_ not available. Signal intensity in vitro decreased with the Fe concentration increase. Tumor darkening observed in vivo.	[155]
Nanospheres	Polymer (PCL)-based composite microsphere with Fe_3_O_4_ nanoparticles and DOX.	DOX	r_2_ = 7.3 mg mL^−1^s^−1^	[156]
Nanospheres	PLGA nanoparticles with oleate-covered iron oxide particles and PTX for AMF heat-induced drug release.	PTX	r_2_ not available; However, 1/T_2_ rates increased gradually vs. the SPION concentrations (in µg/mL of Fe).	[157]
Nanocapsules	Biodegradable and photoluminescent polyester (BPLP) with the PLGA and SPIONs as a polymeric shell. Functionalized with PEG, providing a hydrophilic surface.	proteins	r_2_ = 263 and 237 mM^−1^s^−1^ for non-PEGylated and PEGylated NCs, respectively.	[158]
Nanocapsules	Hydrophobic magnetic nanoparticles entrapped into the PLGA NPs with the human transferrin receptor-binding peptide T7. The water-insoluble drugs, PTX and CUR, loaded into the hydrophobic core.	PTX, CUR	r_2_ = 281.05 mM^−1^s^−1^	[159]
Nanocapsules	DOX-loaded SPIONs@PDA nanoparticles; the shell of the magnetic NP of crosslinked reducible polydopamine and PEG methyl ether methacrylate, with DOPA moiety as an anchor to immobilize SPIONs.	DOX	r_2_ = 33.53 mM^−1^s^−1^	[160]

^1^ BSA—Bovine serum albumin, CUR—curcumin, DOX—doxorubicin, G—gemcitabine, PTX—Paclitaxel.

**Table 4 nanomaterials-13-02163-t004:** Summary of MRI-traceable polymeric nanocarriers for drug delivery applications with contrast agents based on the CEST mechanism.

Structure	Overview	Model Drug ^1^	MRI Contrasting Properties	Ref
Nanocapsules	Acylamino-containing amphiphilic block copolymer (polyethylene glycol-polyacrylamide-poly acetonitrile, PEG-b-P(AM-co-AN)). New nanomedicine: PEG-PAM-PAN@DOX, based on the copolymer, constructed by nanoprecipitation.	DOX	CEST effect at approximately 0.5 ppm; CEST imaging of NCs at different pH revealed a stronger CEST effect at a weak acid or neutral pH.	[167]
Nanospheres	Porous poly(propylene fumarate) (PPF) scaffolds loaded with DOX; the surface of scaffolds modified with three different contrast agents for MRI: iron oxide, manganese oxide, and protamine sulfate (PS-CEST agent).	DOX	CEST signal used for drug release study. The MTR_asym_ at 1.8 ppm showed an increase with incubation time due to the release of PS from the PPF scaffolds.	[168]
Nanocapsules	PEG_114_-b-PDPA_116_ block copolymers that in physiological pH form micelles, and dissociate in an acidic environment.	-	MTR_asym_ dependent on pH; between pH 5 and 6.5 shows a variable CEST contrast.	[169]

^1^ DOX—doxorubicin.

**Table 5 nanomaterials-13-02163-t005:** ^19^F MRI-traceable polymeric nanocarriers for drug delivery applications.

Structure	Overview	Model Drug ^1^	MRI Contrasting Properties	Ref
Nanocapsules	Encapsulation of PTX into core–shell nanocapsules made of a PLGA-PEG shell and PFOB core.	PTX	MRI images obtained by superposition of the ^1^H and ^19^F images of a CT-26 tumor-bearing mouse.	[171]
Nanocapsules	Fluorinated amphiphile with fluorinated moieties as hydrophobic tails and as a ^19^F MRI agent, and monodisperse PEG as hydrophilic heads. Formulation of ^19^F MRI-traceable liposomes with encapsulated DOX based on the fluorinated amphiphile	DOX	In vitro ^19^F signal intensity evaluation for different concentrations of NCs. In vivo superposition of the ^1^H and ^19^F images.	[172]
Nanocapsules	Peptidic monodisperse PEG with fluorinated L-lysine side chains and a fluorescent N-terminal modified for ^19^F MRI and fluorescence dual-imaging traceable and thermo-responsive DOX delivery.	DOX	In vitro ^19^F signal intensity evaluation for different concentrations of NCs. In vivo superposition of the ^1^H and ^19^F images of mice carrying HepG2 tumor.	[173]
Nanocapsules	ROS-sensitive core–shell NCs of diblock polymer; the hydrophilic block of poly(methyl-2-oxazoline) (PMeOx) formed a shell; the hydrophobic block of poly(2,2-difluoroethylacrylamide) (PDFEA) provided ^19^F-NMR signal.	DOX	In vitro ^19^F signal intensity evaluation for different concentrations of NCs. In vivo superposition of the ^1^H and ^19^F images of rat leg after administration of the polymer.	[174]
Nanospheres	Fluorinated PLGA co-polymers (F-PLGA) containing an increasing number of magnetically equivalent fluorine atoms.	DEX, LEF	^19^F-NMR signal at -72 ppm and -70 ppm; T_1_ values of 537 ms and 625 ms and T_2_ values of 122 ms and 60 ms, for F3-PLGA NPs and F9-PLGA NPs, respectively.	[175]
Nanocapsules	Core–shell nanocapsules formed by LbL technique. Shell is composed of Nafion, the fluorinated ionic polymer, and PLL. The surface modified by the adsorption of pegylated polyanion, PGA.	-	^19^F signal arising from Nafion^®^ polymer exhibited multiple resonance lines with T_2_ values in the range of single milliseconds. In vitro imaging of NCs resulted in SNR = 5 (t_acq_ < 30 min) for ^19^F concentration as low as 1.53 × 10^−2^ mM^19^F g^−1^.	[176,177]
Nanocapsules	The 5-FU loaded nanocapsules. Shell formed with polymers: PLL and PGA. The surface modified with PGA-g-PEG.	5-FU	In vitro ^19^F SNR evaluation for the phantom with NCs. SNR = 10 was achieved in t_acq_ of 8 min for the concentration of 982.73 mg/L 5-FU.	[178]
Nanospheres	Hyperbranched polymer (Hyperbranched Poly(N,N-dimethylacrylamide)) covalently conjugated to a biodegradable oligopeptide with 5-FU.	5-FU	Differentiation between attached and released drug states by ^19^F-NMR. 5-FU release induced a change in ^19^F peak width.	[179]

^1^ 5-FU—5-fluorouracil, DEX—Dexamethasone, DOX—doxorubicin, LEF—leflunomide, PTX—Paclitaxel.

## Data Availability

No new data were created or analyzed in this study. Data sharing is not applicable to this article.

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
