# Peer review of "Contrasting Properties of Polymeric Nanocarriers for MRI-Guided Drug Delivery"

_nanomaterials, 2023, doi:10.3390/nano13152163_

Round 1
Reviewer 1 Report
In general, this is a good review paper giving very comprehensive info about drug-delivering nanoparticles containing contrast agents. However, some contents may be redundant and repeated a few times, for instance, the identifications of nanospheres and nanocapsules, which have been repeated in sections 2.2 and 5. So the authors are suggested to refine and remove some repeated content, making the article more coherent. And adding 3 tables to summarise the contents in sections 4.1 to 4.3 briefly will be the article to be matched by readers easily, instead of the whole table 2. All in all, this review is recommended to be published after proper content reorganisation.
The quality of English is good; keeping some contents simpler may help readers to find specific work and understands more.
Author Response
1. Some contents may be redundant and repeated a few times, for instance, the identifications of nanospheres and nanocapsules, which have been repeated in sections 2.2 and 5. So the authors are suggested to refine and remove some repeated content, making the article more coherent.
Repeated contend has been removed from the Conclusions section (5.).
2. Adding 3 tables to summarise the contents in sections 4.1 to 4.3 briefly will be the article to be matched by readers easily, instead of the whole table 2.
Table 2. is now divided into the separate tables for each type of contrast agent (Tables 2-5).
Reviewer 2 Report
In this manuscript, the authors reviewed the recent scientific reports on polymeric nanocarriers for MRI-guided drug delivery, such as constituents and morphologies, mechanisms, and polymeric nanocarriers. However, after reading this manuscript, there are some questions and suggestion below.
1. For the convenience of readers' reading. It is recommended that the author summarize the relevant literature on polymers used for the preparation of nanocarriers (2.3).
2. It is suggested that the author redraw the graphic abstract.
None
Author Response
- For the convenience of readers' reading. It is recommended that the author summarize the relevant literature on polymers used for the preparation of nanocarriers (2.3).
In the section 2.3. a brief description of polymers that were used for the fabrication of the majority of nanocarriers mentioned in the paper is now provided. The main focus was given to physio-chemical and biological properties relevant for the drug delivery aplication.
2. It is suggested that the author redraw the graphic abstract.
It wasn't specified what should be redrawn in the abstract, therefore only resolution of the image was increased
Reviewer 3 Report
The review of Dr. hab., Assoc. Prof. Węglarz and Dr. inż Łopuszyńska is devoted the use of biocompatible nanomaterials (“nanocarriers”) for drug delivery. This approach is known as “theranostics”. It provides to use a more patient-specific approach through the observation of the distribution of contrast agents that are linked to therapeutics. The review described the recent scientific reports on polymeric nanocarriers for MRI-guided drug delivery. Polymeric nanocarriers are a very broad and versatile group of materials for drug delivery, providing high loading capacities, improved pharmacokinetics, and biocompatibility. The work focused on the contrasting properties of proposed polymeric nanocarriers, which can be categorized into three main groups: polymeric nanocarriers with relaxation type contrast agents, with chemical exchange saturation transfer (CEST) properties, and with direct detection contrast agents based on fluorinated compounds. The importance of this aspect tends to be downplayed, despite its being essential for the successful design of applicable theranostic nanocarriers for image-guided drug delivery. Cytotoxicity and therapeutic effects were also summarized.
The review of Prof. Węglarz is excellent, important work in the field of the application of nanomaterials and instructive to the readers. The work is relevant Biology and Medicines section of Nanomaterials.
I have only one comment and suggestion. The work should have been made more visualized and representative. The presented schemes and images have a low resolution and are clearly not enough. Perhaps it would be worth adding a little more chemistry, adding schemes for the synthesis of some of the presented nanomaterials, as well as contrast and therapeutic agents, since the review includes 127 references and provides important information, but only 4 figures are given in the text.
I recommend to publish this manuscript in the Nanomaterials after minor correction

Author Response
The work should have been made more visualized and representative. The presented schemes and images have a low resolution and are clearly not enough. Perhaps it would be worth adding a little more chemistry, adding schemes for the synthesis of some of the presented nanomaterials, as well as contrast and therapeutic agents, since the review includes 127 references and provides important information, but only 4 figures are given in the text.
Figures: 2, 3, 5 and 6 were added. Original figures are in very hight resolution. May be due to the compression in the document the quality seems to be poor. We believe, in the final version of the article, diplayed on the webpage, the original figures will be used.
Round 2
Reviewer 2 Report
I suggest its publication in this journal after addressing the following comments.
1,In Table 3, reference 156, it is recommended that the author unify the format.“r2 = 7.3 mg/mL/s”.Similarly, "F MRI" or "F-MRI","19 F-NMR"or"19F-NMR",“MR imaging”or“MRI”,etc. There are many non-standard abbreviations in the manuscript. Please carefully review the manuscript to ensure consistency in abbreviations. For example: repetition, position, etc.
2,Please correct the numbers: 2.2.1, 2.2.2,..., 2.3.5.
None
Author Response
1a) In Table 3, reference 156, it is recommended that the author unify the format.“r2 = 7.3 mg/mL/s”
The format of the unit was changed to mg mL-1s-1. Recalculating it to the standard unit (mMs-1) was impossible due to the lack of information about molar concentration of NPs.
1b) Similarly, "F MRI" or "F-MRI","19 F-NMR"or"19F-NMR",“MR imaging”or“MRI”,etc. There are many non-standard abbreviations in the manuscript. Please carefully review the manuscript to ensure consistency in abbreviations. For example: repetition, position, etc.
Abbreviations were unified in the manuscript.
2. Please correct the numbers: 2.2.1, 2.2.2,..., 2.3.5.
Nubers of sections are now corrected.